# Post-transcriptional gene silencing mediated by microRNAs is controlled by nucleoplasmic Sfpq

Silvia Bottini [1,2], Nedra Hamouda-Tekaya[1,2], Raphael Mategot[1,2], Laure-Emmanuelle Zaragosi [3], Stephane Audebert [4], Sabrina Pisano[5], Valerie Grandjean[1,2], Claire Mauduit[1,2,6,7], Mohamed Benahmed[2,8], Pascal Barbry [3], Emanuela Repetto[1,2] & Michele Trabucchi [1,2]

There is a growing body of evidence about the presence and the activity of the miRISC in the nucleus of mammalian cells. Here, we show by quantitative proteomic analysis that Ago2 interacts with the nucleoplasmic protein Sfpq in an RNA-dependent fashion. By a combination of HITS-CLIP and transcriptomic analyses, we demonstrate that Sfpq directly controls the miRNA targeting of a subset of binding sites by local binding. Sfpq modulates miRNA targeting in both nucleoplasm and cytoplasm, indicating a nucleoplasmic commitment of Sfpq-target mRNAs that globally influences miRNA modes of action. Mechanistically, Sfpq binds to a sizeable set of long 3′UTRs forming aggregates to optimize miRNA positioning/recruitment at selected binding sites, including let-7a binding to Lin28A 3′UTR. Our results extend the miRNA-mediated post-transcriptional gene silencing into the nucleoplasm and indicate that an Sfpq-dependent strategy for controlling miRNA activity takes place in cells, contributing to the complexity of miRNA-dependent gene expression control.

[1] INSERM U1065, C3M, Team Control of Gene Expression (10), 151 route de St-Antoine-de-Ginestière, B.P. 2 3194, Nice 06204, France. [2] Université Côte d'Azur, INSERM, C3M, 151 route de St-Antoine-de-Ginestière, B.P. 2 3194, Nice 06204, France. [3] Université Côte d'Azur, CNRS, IPMC, Valbonne, France. [4] CRCM, Marseille Protéomique, Institut Paoli-Calmettes, Aix Marseille University, INSERM, CNRS, 27 bd Leï Roure, BP 30059, Marseille 13273, France. [5] Université Côte d'Azur, CNRS, INSERM, IRCAN, Faculty of Medicine, 28 Av. Valombrose, Nice 06107, France. [6] Université Lyon 1, UFR Médecine Lyon Sud, Lyon F-69921, France. [7] Hospices Civils de Lyon, Hopital Lyon Sud, Laboratoire d'Anatomie et de Cytologie Pathologiques, Pierre-Bénite F-69495, France. [8] Centre Hospitalier Universitaire de Nice, Département de Recherche Clinique et d'Innovation, Nice F-06001, France. Silvia Bottini and Nedra Hamouda-Tekaya contributed equally to this work. Correspondence and requests for materials should be addressed to M.T. (email: mtrabucchi@unice.fr)

MicroRNAs (miRNAs) are functional small RNAs and fundamental components of gene expression programs that regulate many biological processes, including cell proliferation, differentiation, and death[1]. Like other small RNAs, miRNAs can be used as biomarkers for human disorders[2]. miRNAs associate with Argonaute (Ago) proteins, mainly Ago2, to form the miRNA-induced silencing complex (miRISC) and target mRNAs[3]. Although miRNA-dependent silencing has mainly been described in the cytoplasm, a growing body of evidence indicates that miRNAs are also functional in the nucleus[4–6]. Canonically, miRNAs use a sequence of 6–8 nucleotides (nt) at their 5′ end, called the seed region, to block the translation or promote the degradation of target mRNAs[1]. Despite efforts to develop bioinformatics tools to predict miRNA-binding sites, it has been demonstrated that prediction approaches can be misleading, yielding approximately 70% false or negative targets[7]. Such a low prediction efficiency might be improved by considering the activity of RNA-binding proteins that bind to mRNAs to control miRNA targeting[1]. In support of this mechanism, it has been shown that some RNA-binding proteins associate with specific mRNAs and interfere with specific miRNA-binding sites to either inhibit or enhance miRNA targeting[8]. This result leads to the concept of a sequence microenvironment surrounding miRNA-binding sites that plays an important role in regulating miRNA activity[8]. Although some examples of such a mechanism have been described, including those for Hu-Antigen R (HuR)[9] and Deadend I (Dnd1)[10], much remains to be learned about the molecular mechanism(s) underlying the roles of the sequences surrounding miRNA-binding sites, and whether this feature is general of miRNA targeting regulation or is confined to specific cases.

Herein, to explore the RNA dependency of miRNA activity we used a quantitative proteomic analysis to identify RNA-dependent Ago2 interactors. Among the identified RNA-dependent interactors, we focused our investigation on Splicing factor proline/glutamine-rich protein (Sfpq). We found that Sfpq interacts with nucleoplasmic miRISC in different human and mouse cell lines. We demonstrated that Sfpq directly promotes miRNA targeting through local binding, ultimately facilitating miRNA-dependent degradation. Although Sfpq only interacts with miRISC in the nucleoplasm, it appeared to modulate miRNA targeting in both the nucleoplasm and cytoplasm. This result indicated that a nucleoplasmic commitment for Sfpq-target mRNAs globally influences miRNA targeting in both cellular compartments. We found that Sfpq preferentially binds to long 3′ UTRs through long sequences that harbor multiple copies of two distinct Sfpq-binding motifs that we had identified. Observations by atomic force microscope (AFM) further showed that Sfpq aggregates onto target 3′UTRs. This process ultimately leads to the position/recruitment optimization of miRNAs at specific binding sites, including let-7a targeting of the Lin28A 3′UTR. In stem cells, Sfpq regulated the let-7-dependent gene expression program toward a neuron-like phenotype differentiation. Our results unveil an unanticipated role for Sfpq in post-transcriptionally promoting miRNA-dependent gene silencing from the nucleoplasm to the cytoplasm of a sizeable subset of mRNAs that have long 3′UTRs. These results highlight the importance of nuclear miRNA targeting and the sequence features of mRNAs for post-transcriptional miRNA programs during gene expression regulation.

## Results

### Identification of RNA-dependent Ago2 interactors. To identify RNA-dependent Ago2 interactors, we performed a label-free quantitative mass spectrometry (MS) analysis[11] of immunoprecipitated Ago2-containing complexes obtained from mouse RAW 264.7 cell extracts that were either undigested ((–) RNase) or totally digested ((+)RNase) (Fig. 1a). Immunoprecipitation (IP) was revealed by SDS-PAGE-silver staining or an RNA gel (Fig. 1b, c). MS analysis identified 915 proteins that might interact with Ago2 (Supplementary Data 1). For quantitative analysis, we applied different filters based on the following criteria: (i) the reproducibility of three replicates using a PCA-based procedure; (ii) the application of a stringent cutoff point for the Mascot score to identify unique peptides; and (iii) the enrichment of proteins identified (abundance score) in the Ago2 immunoprecipitated samples with respect to the IgG IP (ANOVA test, $p$-value ≤ 0.05; Supplementary Data 2). Together, these filters enabled the removal of background noise and resulted in the identification of 299 different proteins in both conditions. After statistical analysis, we identified 133 Ago2 RNA-independent and 166 RNA-dependent interactors (Fig. 1d, Supplementary Fig. 1a, b, and Supplementary Data 3).

Many of the Ago2 interactors found in this study were previously independently identified as being associated with Ago2. Indeed, among the RNA-independent interactors, we identified many miRISC components, including members of the GW protein family (Trinucleotide Repeat Containing 6—TNRC6—in mammals), and CCR4-NOT Transcription Complex Subunit 1 (CNOT1), which, together with DEAD-Box Helicase 6 (DDX6, an RNA-dependent interactor), regulates miRNA-dependent deadenylation of target mRNAs[12]. As expected, we also found several known RNA-dependent interactors, such as Polyadenylate-Binding Protein 1 (PABP1)[13], Elav1 (also called HuR)[14], the putative DExD box helicase Moloney Leukemia Virus 10 (MOV10)[13], and La Ribonucleoprotein Domain Family Member 1 (Larp1)[15]. Notably, although it has been shown that Ago2 also associates with proteins involved in miRNA biogenesis[15], we did not find Dicer, Trans-Activation Responsive RNA-Binding Protein 2 (Trbp2), or Heat Shock Protein 90 (Hsp90) in our analysis, suggesting that the antibody we used specifically immunoprecipitated Ago2 associated with mature miRNAs and not precursors (Supplementary Fig. 1c, d). Taken together, despite some differences, our list of 299 Ago2 interactors is similar to that reported for a previous SILAC quantitative proteomic analysis[15]. Therefore, our results corroborate previous works, indicating the high specificity of our analysis, and extend them by identifying novel interactors.

To study the biological and mechanistic relevance of RNA as a mediator of the interactions between Ago2 and other proteins for miRNA function, we decided to focus on Sfpq, Paraspeckle Component 1 (Pspc1), and Non-POU Domain Containing, Octamer-Binding (NonO), which form a protein complex[16]. These three proteins were among the most abundant RNA-dependent Ago2 interactors (red points in Fig. 1d). All three proteins contain two RNA recognition motif (RRM) domains that confer binding specificity to RNA[17] and a coiled-coil domain that mediates protein aggregation[18]. Sfpq and NonO are multifunctional RNA-binding proteins that can regulate different steps of the mRNA life cycle, including splicing, nuclear localization, and degradation[19, 20]. Pspc1 function is poorly understood. Interestingly, these proteins are known to associate with the long non-coding RNA Nuclear Paraspeckle Assembly Transcript 1 (Neat1) to form the paraspeckle, a nucleoplasmic compartment of approximately 0.2–1 μm in size with several physiopathological functions[16], including nuclear retention of mRNAs[21], transcriptional regulation[22], cancer pathogenesis[23], and viral infection[24]. Interactions with Ago2 were also observed for Sfpq in previous MS analyses in HEK293T cells and in hepatocellular carcinoma[15, 25]. To validate the specific associations between Ago2 and Sfpq, Pspc1, and NonO, we performed co-IP

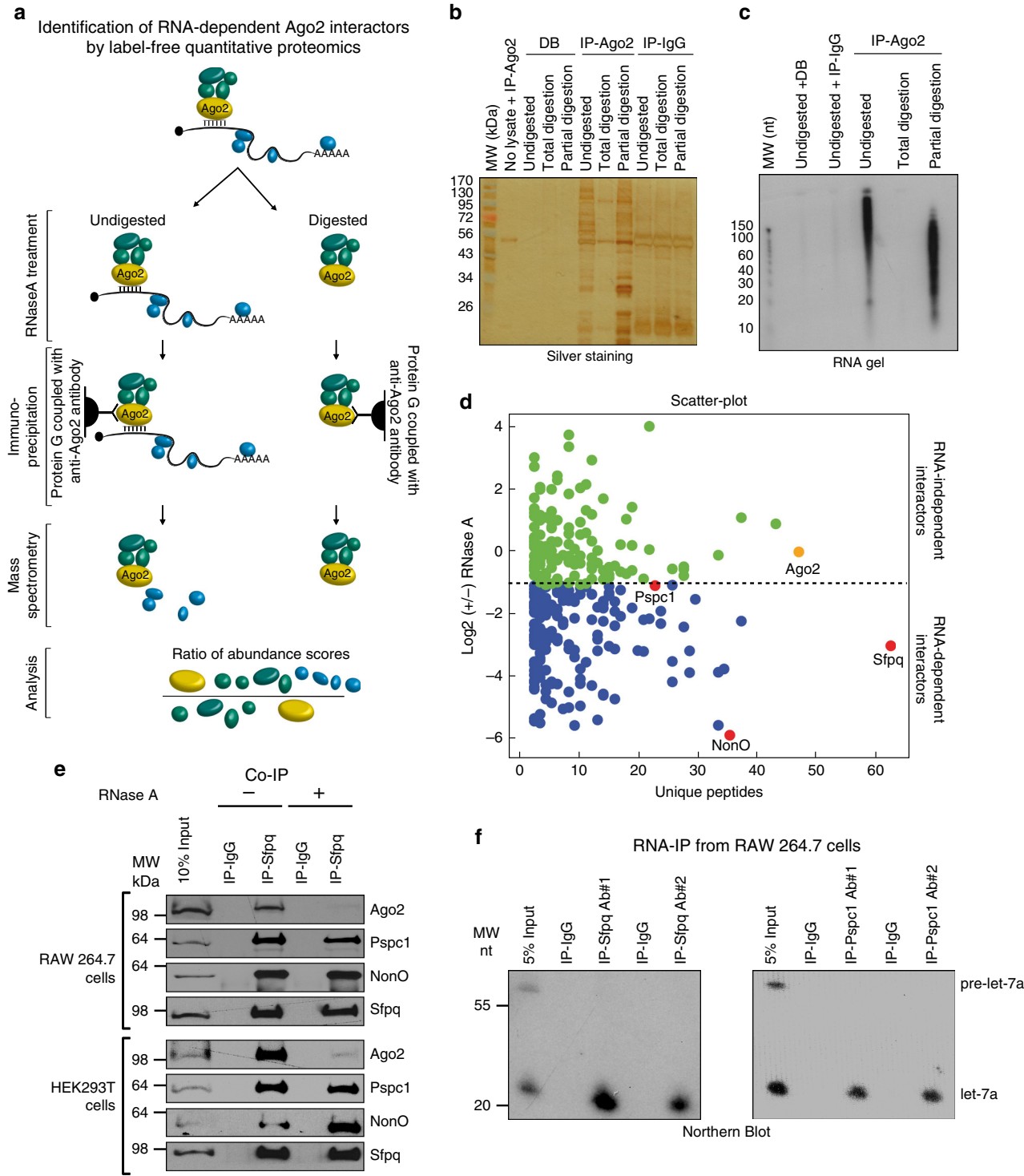

**Fig. 1** Sfpq, Pspc1, and NonO are components of Ago2 complex and interact with let-7a. **a** Overview of the proteomic method used to identify RNA-dependent proteins interacting with Ago2. **b** Silver staining of an SDS-PAGE analysis from the IP with the anti-Ago2 antibody, the protein G conjugated with the Dynabeads (DB), or the anti-Ago2 antibody alone incubated without cell lysate. The samples were untreated, treated with 10 μg ml$^{-1}$ RNase A for 30 min at room temperature (for partial digestion), or treated with 10 mg ml$^{-1}$ RNase A for 30 min at room temperature (for total digestion). **c** Radioactive images of a TBE-Urea gel showing signal from $^{32}$P-labeled RNA fragments of samples untreated, treated with 10 μg ml$^{-1}$ RNase A for 30 min at room temperature (for partial digestion), or treated with 10 mg ml$^{-1}$ RNase A for 30 min at room temperature (for total digestion). **d** Scatter-plot of the log base 2 (−/+) RNase A ratios (abundance scores) plotted with the unique peptides for each identified protein. Each spot is a different protein. **e** Co-IP of endogenous Sfpq and Ago2, Pspc1, or NonO in RAW 264.7 and HEK293T cells. When indicated, cell lysates were incubated at room temperature with RNase A (10 mg ml$^{-1}$) for 30 min. **f** Sfpq and Pspc1 interact with mature let-7a but not with the precursor. RAW 264.7 cell extracts were immunoprecipitated with two different antibodies for each indicated protein. The RNA was purified and analyzed by Northern blotting

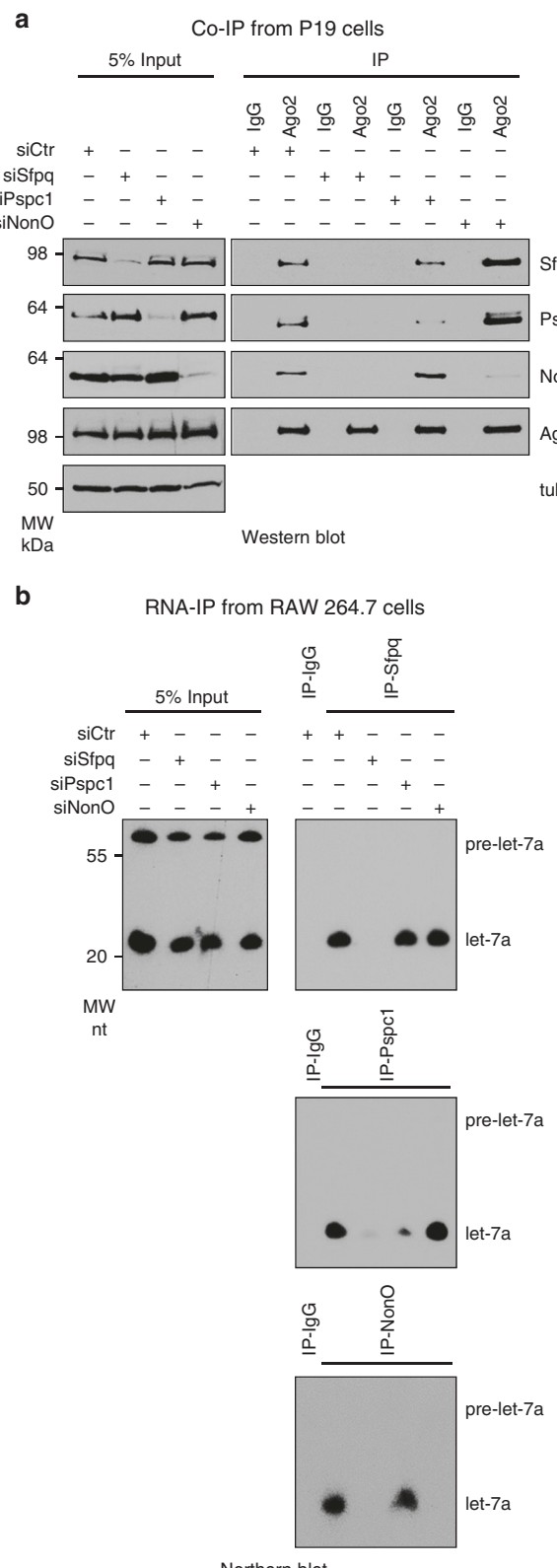

**Fig. 2** Sfpq mediates the interaction between miRISC and Pspc1 or NonO. **a** Co-IP of endogenous Ago2 with Sfpq, Pspc1, or NonO from P19 cells. Cells were transfected with the indicated siRNAs and analyzed by Western blotting. Ago2 and tubulin served as controls. **b** RNA-IP of let-7a with the endogenous Sfpq, Pspc1, or NonO. RAW 264.7 cells were transfected with the indicated siRNAs. Cell extracts were immunoprecipitated with the indicated antibodies and the RNA was purified and analyzed by Northern blotting

experiments with the endogenous proteins in different human and mouse cell lines, including RAW 264.7, P19, and HEK293T cells (Fig. 1e and Supplementary Fig. 2a). In all three cellular models, we confirmed each protein as an integral component of the Ago2 complex. These results were also validated by co-immunoprecipitating Sfpq-tagged and Ago2-tagged proteins or by GST pull-down experiments (Supplementary Fig. 2b, c). RNase treatment confirmed the RNA dependency of the Ago2 interactions; however, it did not affect the interactions among Sfpq, Pspc1, and NonO (Fig. 1e and Supplementary Fig. 2a). To determine whether they also associate with miRNAs, we performed an ribonucleoprotein complexes immunoprecipitation (RNA-IP) followed by Northern blotting for selected miRNAs in different cell lines. Sfpq and Pspc1 interact with let-7a and miR-23b (Fig. 1f and Supplementary Fig. 2d).

Overall, these results validated our MS analysis and raised the possibility that the Sfpq–Pspc1–NonO complex could regulate miRNA functions in an RNA-dependent fashion.

**Nucleoplasmic Ago2 interacts with Sfpq and its complex**. Because Sfpq, Pspc1, and NonO form a complex and share the ability to directly bind single-stranded RNAs, we determined whether one of them specifically mediated the interaction with miRISC. Thus, we singularly knocked down each protein in mouse and human cells and checked for interactions with Ago2 by co-IP and with miRNAs by RNA-IP. As shown in Fig. 2a and Supplementary Fig. 3a and b, Sfpq knockdown specifically inhibited the interaction between Ago2 and either Pspc1 or NonO, whereas Pspc1 or NonO knockdown did not affect the interaction between Sfpq and Ago2. Consistently, Sfpq knockdown also inhibited the interaction between let-7a and either Pspc1 or NonO, whereas Pspc1 or NonO knockdown did not affect the interaction between Sfpq and let-7a (Fig. 2b and Supplementary Fig. 3c).

miRNAs are bound to nuclear Ago2, indicating the existence of a nuclear miRNA pathway[6]. Interestingly, we found that nuclear miRISC co-localizes with Sfpq, Pspc1, and NonO in the nucleoplasm in both human and mouse cell lines by Western and Northern blot analyses (Fig. 3a and Supplementary Fig. 4a). Protein markers for the cytoplasm and the endoplasmic reticulum were absent from nuclear preparations (Fig. 3a). As a second method for testing the nuclear localization of Sfpq and Pspc1, we used co-immunostaining, which clearly indicated that Sfpq and Pspc1 localize in the nucleus, whereas Ago2 is localized in both compartments (Fig. 3b). Co-IP experiments confirmed that Ago2 interacts with Sfpq–Pspc1–NonO complex in the nucleoplasm, but not in the cytoplasm or on chromatin (Fig. 3c and Supplementary Fig. 4b, c). Together, our results indicate that Sfpq mediates the interaction of both Pspc1 and NonO with nucleoplasmic miRISC. Therefore, Sfpq appears to mediate the sequence specificity of this complex to modulate nucleoplasmic miRISC activity.

**Sfpq promotes miRNA binding on a subset of binding sites**. To investigate whether Sfpq associates to all or to a subset of miR-NAs, we performed an Sfpq RNA-IP experiment in RAW 264.7 cells followed by high-throughput small RNA sequencing analysis and compared the results with the Ago2 RNA-IP-sequencing data (Supplementary Data 4). We found that 224 and 259 mature miRNAs associated with Sfpq and Ago2, respectively (Supplementary Data 4 and Supplementary Fig. 5a). All 224 miRNAs enriched in the Sfpq IP were also associated with Ago2 (Fig. 4a). Most of the 35 remaining miRNAs that associated solely with Ago2 were miRNA passenger strands (Supplementary Data 4).

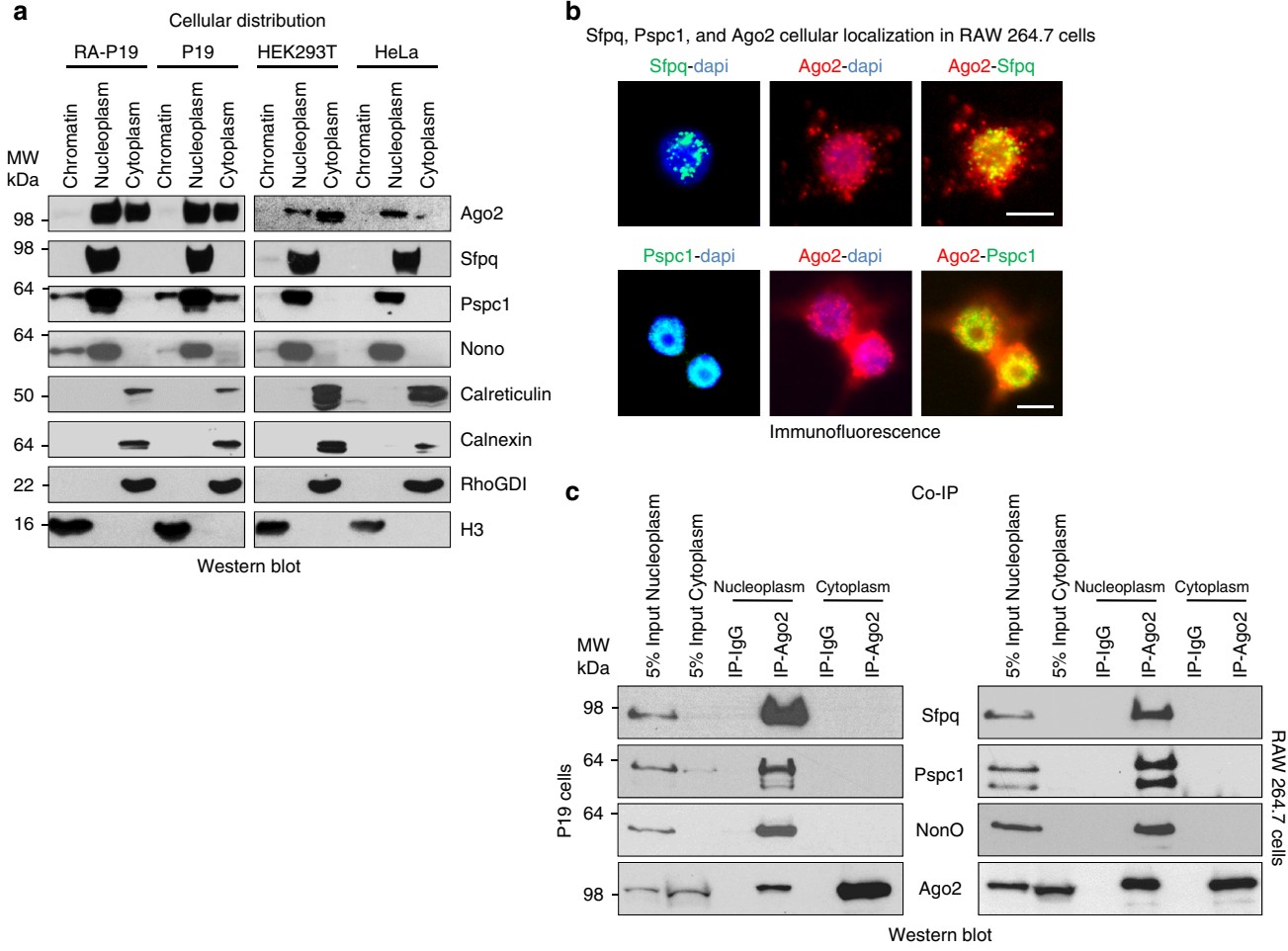

**Fig. 3** Sfpq, Pspc1, and NonO interact with Ago2 in the nucleoplasm. **a** Immunoblot analysis of chromatin, nucleoplasm, or cytoplasm from HeLa, HEK293T, P19, or RA-treated P19 cells using antibodies directed to the indicated proteins. **b** Co-immunofluorescence analysis of RAW 264.7 cells stained with the indicated antibodies. Scale bar corresponds to 10 μm. **c** Co-IP of endogenous Ago2 and the indicated proteins from nucleoplasmic or cytoplasmic extracts from RAW 264.7 or P19 cells

Therefore, we concluded that all the miRNAs associated with Sfpq are functionally active, given that they also associate with Ago2. These miRNAs include the let-7 family, miR-21, miR-31, miR-106b, and miR-23b. To validate this analysis, RNA-IP followed by either Northern blot or quantitative reverse-transcriptase PCR (RT-qPCR) analyses in both RAW 264.7 and HEK293T cells confirmed that let-7a, miR-23b, miR-125b, and miR-24 associate with Sfpq (Fig. 1f and Supplementary Figs. 2d, 5b). Notably, Sfpq only co-immunoprecipitated with mature miRNAs, not with precursors. These results indicate that Sfpq may be involved in the miRNA mode of action but not in miRNA biogenesis. Indeed, Sfpq knockdown did not affect miRNA expression or Ago2-loaded miRNA levels (Fig. 4b). Therefore, these results prompted us to investigate the possible role of Sfpq in regulating miRNA activity in gene silencing.

We hypothesized that similar to other RNA-binding proteins, Sfpq could modulate miRNA-binding activity. To test this hypothesis, we performed HIgh-Throughput Sequencing of RNA isolated by CrossLinking IP (HITS-CLIP) on Ago2 in the presence or absence of Sfpq. We performed this analysis on ectopically expressed let-7a and endogenous miRNAs in stem cells. We selected let-7a as a model for the following reasons: (i) let-7 is the most associated miRNA with Sfpq (Supplementary Data 4); (ii) ingenuity pathway analysis from the Sfpq RNA-IP experiment followed by small RNA sequencing showed an

enrichment of cell cycle and cell development biological processes (Supplementary Fig. 5c), and let-7 controls both cell cycle and cell development[26]; (iii) both Sfpq and let-7 control nervous system development[27, 28]; and (iv) let-7 has been largely used as a model to study miRNA biology and biochemistry[29]. Although not expressed in stem cells, let-7 expression is induced upon retinoic acid (RA) stimulation to promote cell cycle arrest and differentiation[30]. As shown in Fig. 4c, Sfpq, Pspc1, and NonO co-immunoprecipitated with mature let-7a in RA-stimulated P19 cells, a cellular model for embryonic stem cells. Therefore, to investigate the impact of Sfpq on the modulation of let-7a-binding activity, we conducted HITS-CLIP mapping of Ago2 before and after Sfpq knockdown in let-7a-transfected P19 and control cells (Supplementary Data 5). Levels of ectopic Ago2-loaded let-7a in transfected P19 cells were comparable to that of endogenous miRNAs (Supplementary Fig. 5d), and its cellular localization was similar to that of RA-treated P19 cells (Supplementary Figs. 4a, 5e). Briefly, HITS-CLIP analysis identified 4202 Ago2-peaks specific for the let-7a condition (Ago2-let-7a peaks), which were distributed in different parts of the transcriptome, including the 3′UTR and protein coding sequence (CDS), as the main target substrates of let-7a (Supplementary Fig. 6a and Supplementary Data 5)[1]. To assess whether Sfpq directly regulates the identified Ago2-let-7a peaks, we first mapped the Sfpq-binding sites by comparing the Sfpq

HITS-CLIP in siSfpq-transfected P19 cells with sicontrol-transfected cells. Sfpq peaks identified by HITS-CLIP mainly co-localized with the Ago2 peaks in the 3′UTR (Supplementary Data 5 and Supplementary Fig. 6b). Hence, we calculated the distance between the Ago2-let-7a peaks and the closest Sfpq peaks. All the Ago2-let-7a peaks were then grouped into the

following three categories: (1) close to Sfpq peaks, within a distance of 500 nt; (2) far from Sfpq peaks, between a distance of 500–7000 nt; and (3) very far from Sfpq peaks, with a distance of 7000 nt or more (virtually not bound to Sfpq). We reasoned that the close distance would provide information about the local and direct roles of Sfpq in modulating Ago2-binding activity, whereas

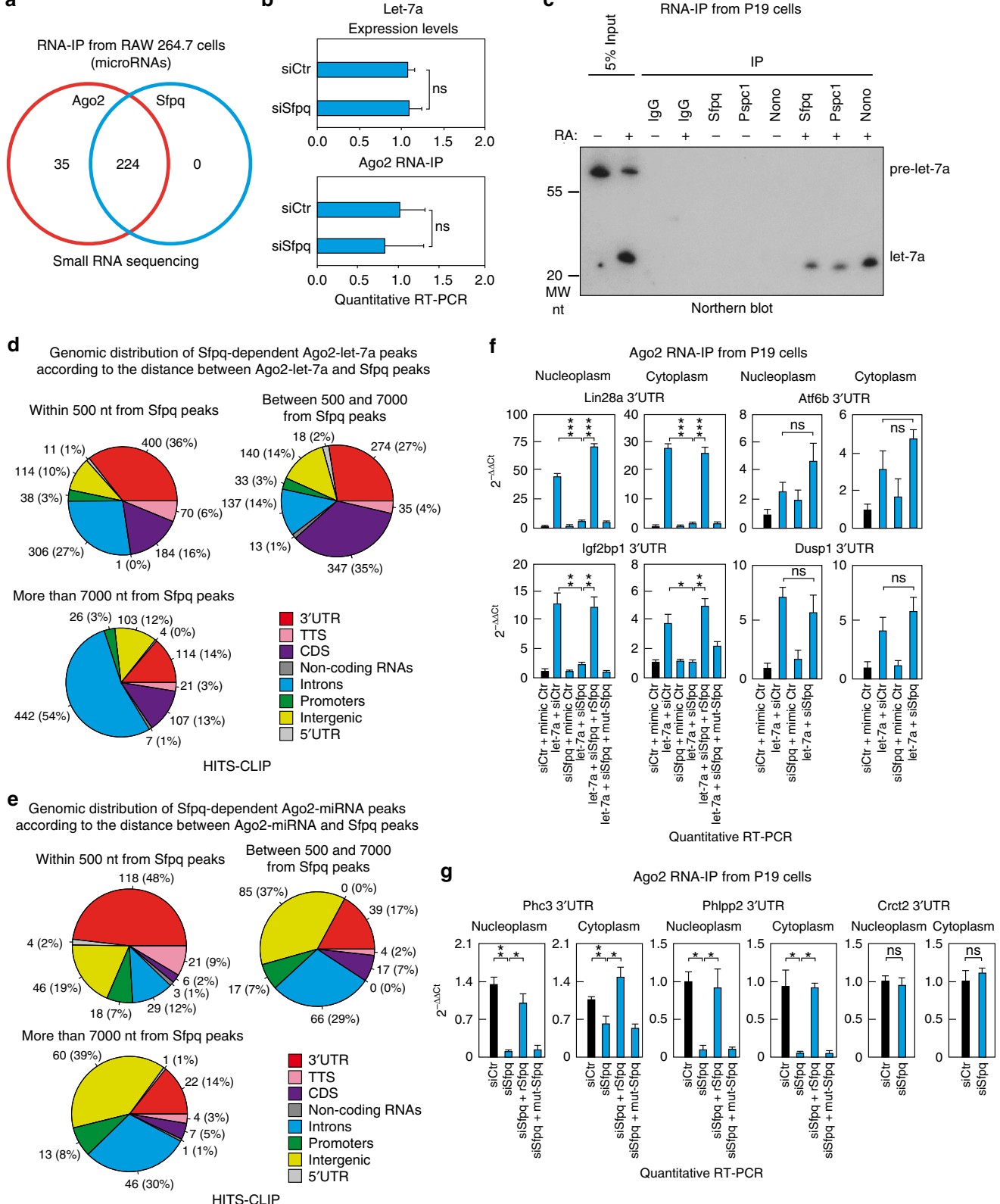

far and very far distances would represent indirect effects of Sfpq knockdown, providing important internal controls for our analyses. Using this approach, we identified 1134 Ago2-let-7a peaks that were reduced in the absence of Sfpq, with a close distance to an Sfpq peak (Supplementary Fig. 6c). Among them, only those that mapped in the 3′UTR were significantly enriched compared with those with far or very far distances from Sfpq peaks (Fig. 4d; Fischer exact test $p$-value = 2.2E-16 for both comparisons). To determine whether 500 nt is the critical distance to define a direct regulation of Ago2-binding activity by Sfpq, we reasoned that within this distance, but not further, we should observe a significant reduction in Ago2-let-7a peaks upon Sfpq knockdown compared to a random distribution of the distance calculated computationally. Briefly, we divided the distance between Sfpq peaks and the reduced Ago2-let-7a peaks upon Sfpq knockdown from 0 to 1000 nt in bins of 250 nt each in the 3′UTRs. Then, we computationally shuffled 10,000 times the relative position of Sfpq and the reduced Ago2-let-7a peaks upon Sfpq knockdown in each bin and calculated the $Z$-score as a statistical test (Supplementary Fig. 6d). This analysis indicated that (i) at the closest distances between Sfpq and the Sfpq-dependent Ago2-let-7a peaks, the functional connections between Ago2 and Sfpq are greater and (ii) that this functional connection is significant up to a distance of 500 nt. Therefore, these data suggest that 500 nt is the maximal distance that can be considered to likely indicate a direct regulation of Ago2-binding activity by Sfpq. Importantly, among the peaks that are directly controlled by Sfpq, we found different let-7a-binding sites to those already validated in the mRNAs encoding for Lin28A[31], Hic2[32], Mier2, and Igf2bp1[33]. Similarly, we found that the direct reduction of the binding sites for the endogenously expressed miRNAs (endogenous Ago2-miRNA peaks; Supplementary Fig. 6e)[34] in P19 cells upon Sfpq knockdown statistically only occurred in the 3′UTR (Fig. 4e and Supplementary Fig. 6f; Fischer exact test, $p$-values = 3.18E-13 and 9E-13 for the close vs. far distance and close vs. very far distance comparisons, respectively). Importantly, we found that 24 and 8.2% of Sfpq-independent Ago2-let-7a peaks and endogenous Ago2-miRNA peaks, respectively, are located in the 3′UTR within a close distance to Sfpq peaks (<500 nt), suggesting that only a subset of miRNA-binding sites is directly controlled by Sfpq. To facilitate the accessibility and correct interpretation of all our HITS-CLIP results, we created an online database with a user-friendly interface containing the genomic coordinates for both the Ago2 and Sfpq peaks in each category we considered (http://trabucchilab.unice.fr/SITO/index.php#). Together, this analysis indicates that the 3′UTR is the substrate by which Sfpq can directly promote miRNA targeting at selected binding sites.

To validate these results, we performed Ago2 RNA-IP experiments followed by RT-qPCR. P19 cells were transfected with let-7a and/or with a different siSfpq from those used for the HITS-CLIP experiment (Supplementary Fig. 6g). We separated the nucleoplasm from the cytoplasm, partially digested the RNA and performed RNA-IP followed by RT-qPCR from each compartment. Interestingly, Sfpq knockdown impaired the activity of let-7a-binding sites on the Lin28A and Igf2bp1 3′ UTRs in both the nucleoplasm and cytoplasm, but not on the Atf6b and Dusp1 3′UTRs, which were controls for Sfpq-independent let-7a-binding sites (Fig. 4f). By incubating the lysates from the siSfpq-let-7a-transfected cells with the recombinant wild-type Sfpq before RNA digestion, we were able to rescue this effect (Fig. 4f). The recombinant Sfpq mutant with L535, L539, L546, and M549 substituted to alanine was unable to rescue the miRNA targeting. The alanine mutations were designed to disrupt Sfpq ability to aggregate but not its binding activity, which was previously reported[18]. These data indicate that Sfpq binding and aggregation directly promote miRNA targeting at selected binding sites. Similar data were also obtained in human NTERA-2 stem cells and for the endogenous stem cell-specific miR-302b[35, 36] in both P19 and NTERA-2 cells (Fig. 4g and Supplementary Fig. 6h). Therefore, these data indicate that Sfpq directly controls a subset of miRNA-binding sites through local binding. Despite its nucleoplasmic localization, Sfpq enhances miRNA-binding activity in both the nucleoplasm and cytoplasm, suggesting that it promotes a nucleoplasmic commitment of mRNAs to globally control miRNA targeting.

**Sfpq controls mRNA silencing by specific miRNA-binding sites.** To assess whether Sfpq-dependent miRNA targeting correlates with changes in target mRNA gene expression levels, we analyzed the mRNA expression profile in control and let-7a-transfected P19 cells upon Sfpq knockdown (Supplementary Data 6). Consistent with our HITS-CLIP data, Sfpq knockdown significantly rescued the magnitude of downregulation of those transcripts containing direct Sfpq-dependent Ago2-let-7a peaks in the 3′UTR (Supplementary Fig. 7a; Wilcoxon test, $p$-value = 4E-7), but not for other transcripts containing Sfpq-indirect (far or very far distances between Ago2-let-7a and Sfpq peaks) or Sfpq-independent Ago2-let-7a peaks (Supplementary Fig. 7b). Because canonical miRNA-binding sites confer a more potent downregulation[37], we analyzed the rescue of the silencing for the 12 Ago2-let-7a peaks that contain canonical let-7a-binding sites in the 3′UTR and whose binding is directly promoted by Sfpq, namely, Lin28A, Igf2bp1, Hic2, Mier2, Notch2, Map1b, Bbx, Skil, Lamp2, Tmem194, Ash1l, and Gns. As shown in Fig. 5a, in this case, the let-7a-mediated silencing and the rescue upon Sfpq knockdown was more striking than the full set of direct Sfpq-dependent target 3′UTRs (Wilcoxon test, $p$-value = 4E-6). This analysis was validated by performing RT-qPCR analyses on six of the selected let-7a target mRNAs, including four Sfpq-dependent and two Sfpq-independent targets, namely, Lin28A, Hic2, Mier2, Igf2bp1, Atf6b, and Dusp1 (Supplementary Fig. 7c). This

**Fig. 4** Sfpq promotes miRNA targeting at selected binding sites. **a** Venn diagram of Ago2 or Sfpq RNA-IP-enriched miRNAs found by small RNA sequencing analysis. **b** Let-7a expression levels (upper panel) and RNA-IP of Ago2 and let-7a (lower panel) in control or siSfpq-transfected RAW 264.7 cells. RNA extracts were analyzed by RT-qPCR. Data are presented as the mean ± s.e.m. ($n = 6$) and normalized to U2 snRNA or the input, respectively. **c** Sfpq, Pspc1, and NonO interact with mature let-7a in RA-treated P19 cells. Cell extracts were immunoprecipitated with the indicated antibodies and RNA was purified and analyzed by Northern blotting. Genomic distribution of either the **d** Ago2-let-7a or **e** endogenous Ago2-miRNA peaks decreased upon Sfpq knockdown, according to the distance to Sfpq peaks by HITS-CLIP analysis. These data show the prevalence of the Sfpq-dependent Ago2 peaks in the 3′ UTR when Sfpq binds closely (<500 nt). RNA-IP of Ago2 and the indicated 3′UTRs for either **f** let-7a or **g** the endogenously expressed miR-302b-binding sites. P19 cells were transfected with the indicated molecules. Nucleoplasm and cytoplasm fractions were separated. The indicated cell lysates were incubated with either 100 nM full-length recombinant wild-type Sfpq or the Sfpq-214–598 quadruple mutant (L535A, L539A, L546A, and M549A) for 30 min at room temperature. Before IP, the lysate was partially digested with 10 μg ml⁻¹ RNase A for 30 min at room temperature. RNA was purified from the immunocomplexes and from 5% of the input and analyzed by RT-qPCR using oligonucleotide probes surrounding the miRNA-binding sites identified by HITS-CLIP. Data are presented as the mean ± s.e.m. ($n = 3$) and normalized to their own inputs. Student's $t$-test (for **b**) or one-way ANOVA followed by Tukey's post hoc test (for **f**, **g**) with *$p < 0.05$ and **$p < 0.01$. *ns* not significant, *TTS* transcription termination site

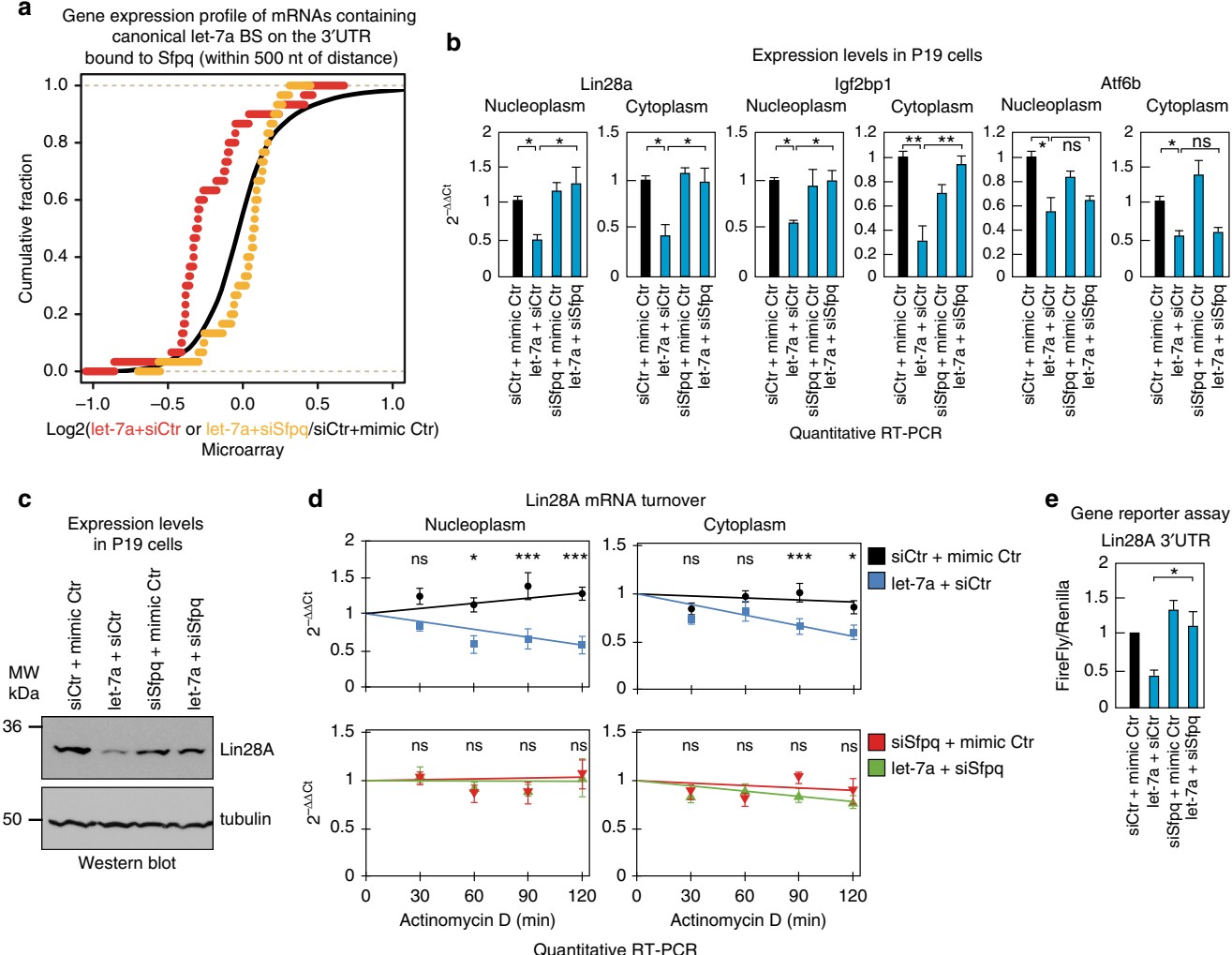

**Fig. 5** Sfpq promotes post-transcriptional silencing mediated by miRNAs in both nucleoplasm and cytoplasm. **a** Gene expression differences in let-7a-transfected P19 cells upon Sfpq knockdown and control. Differential expression is plotted for mRNAs containing canonical let-7a-binding sites (BS) localized in the 3′UTR within a close distance to Sfpq peaks (<500 nt) and reduced upon Sfpq knockdown. **b** P19 cells were transfected with the indicated molecules. Nucleoplasm and cytoplasm were separated to measure the expression levels of the indicated let-7a-target mRNAs. RNA was purified and analyzed by RT-qPCR. Data are normalized with U2 snRNA and presented as the mean ± s.e.m. (n = 3). **c** Immunoblot analysis of Lin28A and tubulin in P19 cells transfected with let-7a and/or siSfpq. **d** Quantitative RT-PCR analysis of the half-life of Lin28A transcript in let-7a-transfected P19 cells compared to control (upper panels) or siSfpq + let-7a-transfected cells compared to siSfpq control (lower panels). Total RNA from either nucleoplasm (left panels) or cytoplasm (right panels) was isolated at the indicated times after addition of actinomycin D. Data are normalized with β2-microglobulin and presented as the mean ± s.e.m. (n = 6). **e** Relative luciferase activity of reporter constructs containing the mouse Lin28A 3′UTR sequence in HEK293T cells transfected with let-7a and siSfpq as indicated. The data were normalized using Renilla activity and presented as the mean ± s.e.m. (n = 4). One-way ANOVA followed by Tukey's post hoc test: *$p < 0.05$, **$p < 0.01$, ns not significant

validation was performed with a different siRNA from that used for the transcriptomic analysis. Similarly, Sfpq knockdown significantly upregulated the steady-state expression of those transcripts that contain direct Sfpq-dependent Ago2 peaks with canonical binding sites for the 20 most expressed endogenous miRNAs in the 3′UTR (Supplementary Fig. 7d; Wilcoxon test, p-value = 0.04769), but not that of the other transcripts containing indirect Sfpq-independent Ago2 peaks with canonical miRNA-binding sites (Wilcoxon test, p-value = 0.07729 for far and 0.9939 for very far distances, respectively; Supplementary Fig. 7d). However, no significant upregulation was observed upon Sfpq knockdown when we considered the whole set of mRNAs with endogenous Ago2 peaks containing canonical or non-canonical miRNA-binding sites (Supplementary Fig. 7e). The absence of any siSfpq effects on the whole data set of endogenous Ago2

peaks could be due to the very mild downregulation conferred by endogenous non-canonical miRNA-binding sites[37].

As shown in Fig. 5b and Supplementary Fig. 8a, the downregulation of selected direct Sfpq-dependent mRNAs targeted by let-7a or by the endogenously expressed miR-302b was rescued in both the nucleoplasm and cytoplasm by Sfpq knockdown, but not the Sfpq-independent miRNA-target mRNA control Atf6b mRNA. Similar results were also obtained in human NTERA-2 cells (Supplementary Fig. 8b). This Sfpq-dependent mechanism of miRNA targeting controls let-7a-dependent differentiation programs in stem cells (Supplementary Fig. 8c). Therefore, these data support the hypothesis that Sfpq plays a role in regulating miRNA silencing on specific binding sites, which globally impact miRNA-dependent gene expression programs.

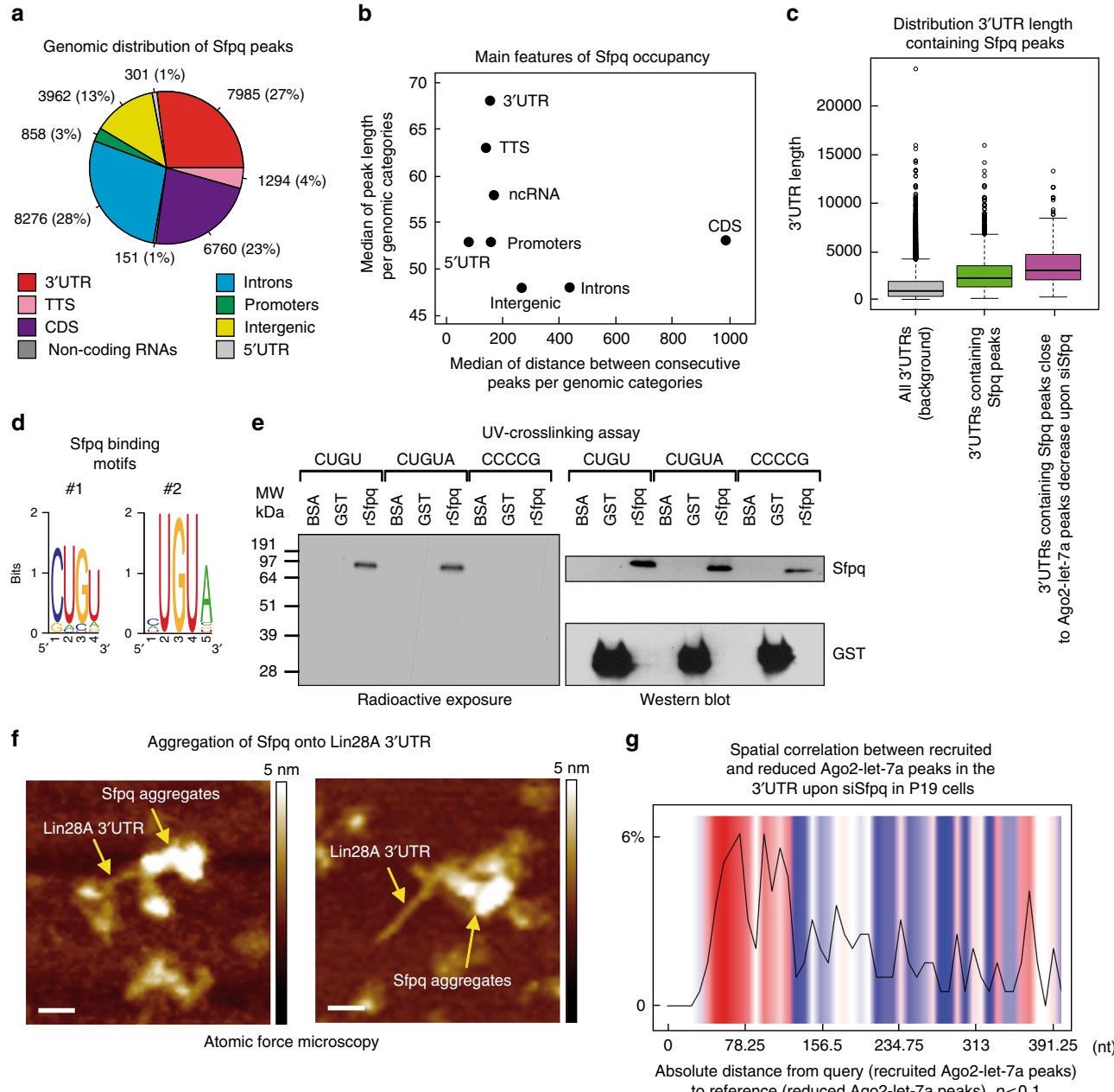

**Fig. 6** Sfpq aggregates onto long 3′UTRs to modulate the accessibility of miRNA-binding sites. **a** Genomic distribution of Sfpq peaks by HITS-CLIP analysis. **b** Scatter plot of length and frequency of Sfpq peaks in different genomic regions. **c** Distribution of the 3′UTR length of the indicated subpopulation of transcripts. **d** Result of the bioinformatic analysis for de novo search of Sfpq-binding motif from Sfpq HITS-CLIP data. **e** UV-crosslinking assay to analyze the interaction of the recombinant Sfpq (100 nM) with the $^{32}$P-labeled RNA oligonucleotides containing two copies of the CUGU or CUGUA, respectively, but not to the CCCCG negative control sequence. **f** Top view of two topographic AFM images of typical Sfpq-Lin28A 3′UTR complexes. The color bar on the right represents the height scale with a maximum corresponding to 5 nm. Scale bar corresponds to 20 nm. **g** Spatial correlation of the recruited Ago2-let-7a peaks compared to the decreased ones. In the x-axis is reported the distance in nt between the reduced and the new recruited peaks on the 3′UTR, whereas in the y-axis is reported the percentage of recruited peaks. The black line is the data density, the red bars indicate that the number of recruited peaks is higher than that expected by chance, the blue bars indicate the opposite, whereas the white bars indicate no recruitment

To further inspect the direct role of Sfpq in controlling miRNA targeting in cells, we focused on a particular known let-7a target —the oncogene Lin28A, also known as *Lin-28 homolog A*, which regulates the self-renewal of stem cells and cancer stem cells[38]. Sfpq knockdown in either P19 or NTERA-2 cells abrogated the let-7a-mediated decrease in the steady-state levels of Lin28A protein in either P19 or NTERA-2 cells (Fig. 5c and Supplementary Fig. 9a). According to a previous work[31], we observed a shorter half-life for endogenous nucleoplasmic and cytoplasmic

Lin28A mRNA in let-7a-transfected P19 cells compared with the control cells. This shortened half-life was rescued by Sfpq knockdown (Fig. 5d). Nuclear run-on (NRO) experiments ruled-out a transcriptional modulation of Lin28A by let-7a or Sfpq (Supplementary Fig. 9b, c). Overall, these data indicate that Sfpq globally modulates the Lin28A mRNA turnover rate through let-7a, resulting in a significant decrease in the steady-state expression levels of the mRNA protein.

To further confirm the direct involvement of Sfpq in controlling let-7a silencing activities on the Lin28A 3′UTR, we used a reporter plasmid in which the Lin28A 3′UTR was cloned downstream of a luciferase open reading frame. Overexpression of let-7a significantly reduced the luciferase activity in HEK293T cells transfected with the reporter plasmid, which was rescued upon Sfpq knockdown (Fig. 5e). However, siSfpq failed to produce any effects on a reporter construct containing only the six let-7a-binding sites (Supplementary Fig. 9d) and on the empty vector (Supplementary Fig. 9e). By contrast, Sfpq knockdown also failed to produce any effects on the miR-125b-mediated downregulation of the Lin28A 3′UTR[39] (Supplementary Fig. 9f), confirming the presence of Sfpq-independent binding sites even with close Sfpq peaks, which was also observed by the HITS-CLIP analysis. Moreover, Sfpq-binding activity on the Lin28A mRNA is independent of let-7a activity (Supplementary Fig. 9g). Overall, these data suggest that Sfpq binding to specific cis-elements on the Lin28A 3′UTR optimizes the positioning/recruitment of miRNAs to selected binding sites.

**Sfpq aggregates on target 3′UTRs to promote miRNA targeting.** To gain insights into the mechanism by which Sfpq regulates specific miRNA-binding sites in the 3′UTR, we investigated the Sfpq HITS-CLIP data in P19 cells to uncover the presence of any peculiar feature(s) of Sfpq-binding activity. Briefly, we found approximately 30,000 Sfpq peaks, with the majority of them mapping in introns and 3′UTRs (Fig. 6a). Interestingly, Sfpq peaks in the 3′UTRs are more elongated and closer to one another than those that map in other genomic regions (Fig. 6b). Transcription termination sites (TTS) and non-coding RNAs show similar trends, whereas Sfpq peaks in the introns show opposite features. Additionally, as shown in Fig. 6c, Sfpq occupancy is enriched in long 3′UTRs compared to the entire 3′UTR data set (Wilcoxon test, $p$-value = 2.2E-16). The 3′UTR length was even longer when we selected the 3′UTRs that contain at least one Sfpq peak and Sfpq-dependent let-7a-binding sites (Wilcoxon test, $p$-value = 2.2E-16 compared to the length of 3′UTRs-containing Sfpq peaks). These data suggest that Sfpq preferentially binds to long 3′UTRs, forming long portions of occupancy.

Next, we looked for specific Sfpq-binding motifs. Because RRM domains bind an average of 4 nt[40], we looked for motifs of 4–6 nt in length in Sfpq peaks from the HITS-CLIP data set. Briefly, we searched for de novo motifs by dividing the Sfpq HITS-CLIP data set into the following three groups: (i) all Sfpq peaks; (ii) Sfpq peaks located in the 3′UTR; and (iii) Sfpq peaks within a close distance (<500 nt) to the Sfpq-dependent Ago2 peaks. In this analysis, we have found enriched motifs with a core composed of the UGU sequence (Supplementary Fig. 10a). Using motif clusterization[41], we obtained two 4- and 5 nt-enriched consensus motifs (Fig. 6d). To validate these two motifs, we used recombinant Sfpq and synthetic RNA in UV-crosslinking and electrophoresis mobility shift (EMSA) assays (Fig. 6e and Supplementary Fig. 10b). BSA, GST, and Ago2, which were used as controls, did not show any binding activity to the newly discovered Sfpq-binding sequences; neither did a negative control sequence with recombinant Sfpq (Fig. 6e and Supplementary Fig. 10b, c). Importantly, these two binding motifs were significantly enriched in the Sfpq peaks located in the 3′UTR that map very close to Sfpq-dependent Ago2-let-7a peaks with respect to all the 3′UTR Sfpq peaks (Fisher exact test, $p$-value = 2.2E-16).

Because Sfpq aggregates to form polymers on nucleic acids[17, 18] to facilitate miRNA targeting (Fig. 4f, g), we hypothesized that the two Sfpq-binding motifs would serve as substrates to promote Sfpq aggregation on target 3′UTRs. To test this hypothesis, we performed Pearson and Spearman correlation tests between the number of occurrences of binding motifs within the peak sequences and the peak length. We found a higher positive correlation in Sfpq peaks that map within 500 nt from Sfpq-dependent Ago2-let-7a peaks and their length (Spearman correlation for moti#1: 0.56; Pearson correlation for motif#1: 0.85; Spearman correlation for motif#2: 0.54; and Pearson correlation for motif#2: 0.79), than in all the Sfpq peaks in the 3′UTR (Spearman correlation for moti#1: 0.55; Pearson correlation for motif#1: 0.72; Spearman correlation for motif#2: 0.48; and Pearson correlation for motif#2: 0.62), or in all the Sfpq peaks (Spearman correlation for moti#1: 0.48; Pearson correlation for motif#1: 0.65; Spearman correlation for motif#2: 0.42; and Pearson correlation for motif#2: 0.58). This analysis indicated that the highest frequency of Sfpq-binding sites occurred close to direct Sfpq-dependent Ago2 peaks, thus supporting the hypothesis that Sfpq has a tendency to form long aggregates that can ultimately promote the positioning/recruitment of miRNAs on selected and close binding sites. In fact, the Lin28A 3′UTR contains 34 Sfpq-binding sites, and recombinant wild-type Sfpq forms aggregates along its sequence, as we demonstrated using AFM (Fig. 6f and Supplementary Fig. 11a), but not the recombinant Sfpq-214–598 quadruple mutant (L535A, L539A, L546A, and M549A), which was unable to aggregate while binding to the RNA (Supplementary Figs. 10c, 11b)[18]. BSA did not bind to Lin28 3′UTR, which is removed from the untreated muscovite mica surface in the washing step (Supplementary Fig. 11c). Furthermore, in support of this conclusion, our HITS-CLIP analyses demonstrated that 70% of the 3′UTRs in which Ago2-let-7a peaks are directly controlled by Sfpq (Sfpq peaks within 500 nt of distance) undergo recruitment of new Ago2 peaks upon Sfpq knockdown, whereas for the endogenous Ago2-miRNA peaks, this value is 98% (Supplementary Data 7). Spatial correlation between the recruited and reduced Ago2 peaks upon Sfpq knockdown indicated that recruitment mainly occurs close to the Sfpq-dependent Ago2 peaks (Fig. 6g). These data suggest that the presence of Sfpq locally modulates miRNA accessibility/positioning to specific binding sites by forming aggregates that may modulate the secondary structure of the target 3′UTR.

Finally, RNA-sequencing analysis of control and siSfpq-transfected P19 cells ruled out any roles for Sfpq in regulating alternative splicing or alternative polyadenylation sites in the 3′ UTRs in which Sfpq directly promotes miRNA targeting (Supplementary Fig. 12a, b, and Supplementary Data 8).

Overall, these data indicate that the presence of specific Sfpq-binding sites determines the fate of a cohort of mRNAs, where Sfpq forms long aggregates in the 3′UTR to modulate 3′UTR folding for the proper positioning/recruitment of miRNAs to selected binding sites, whereas avoiding random binding of miRNAs that would not be effective.

## Discussion

The interplay between miRNAs and RNA-binding proteins has been dubbed "the post-transcriptional regulatory code"; by interacting and competing to binding sites, these post-transcriptional modulators dictate metabolic impact on cognate RNAs to regulate stability, localization, and protein synthesis[8]. Our study added Sfpq to the list of other RNA-binding proteins, including HuR, Pumulio, polypyrimidine tract-binding protein, cytoplasmic polyadenylation element-binding proteins, Dnd1, and RNA-binding motif protein 38 (RBM38), that have been shown to be involved in regulating miRNA targeting in mammalian cells[42]. In comparison with previous studies where specific RNA-binding proteins appear to regulate some cytoplasmic

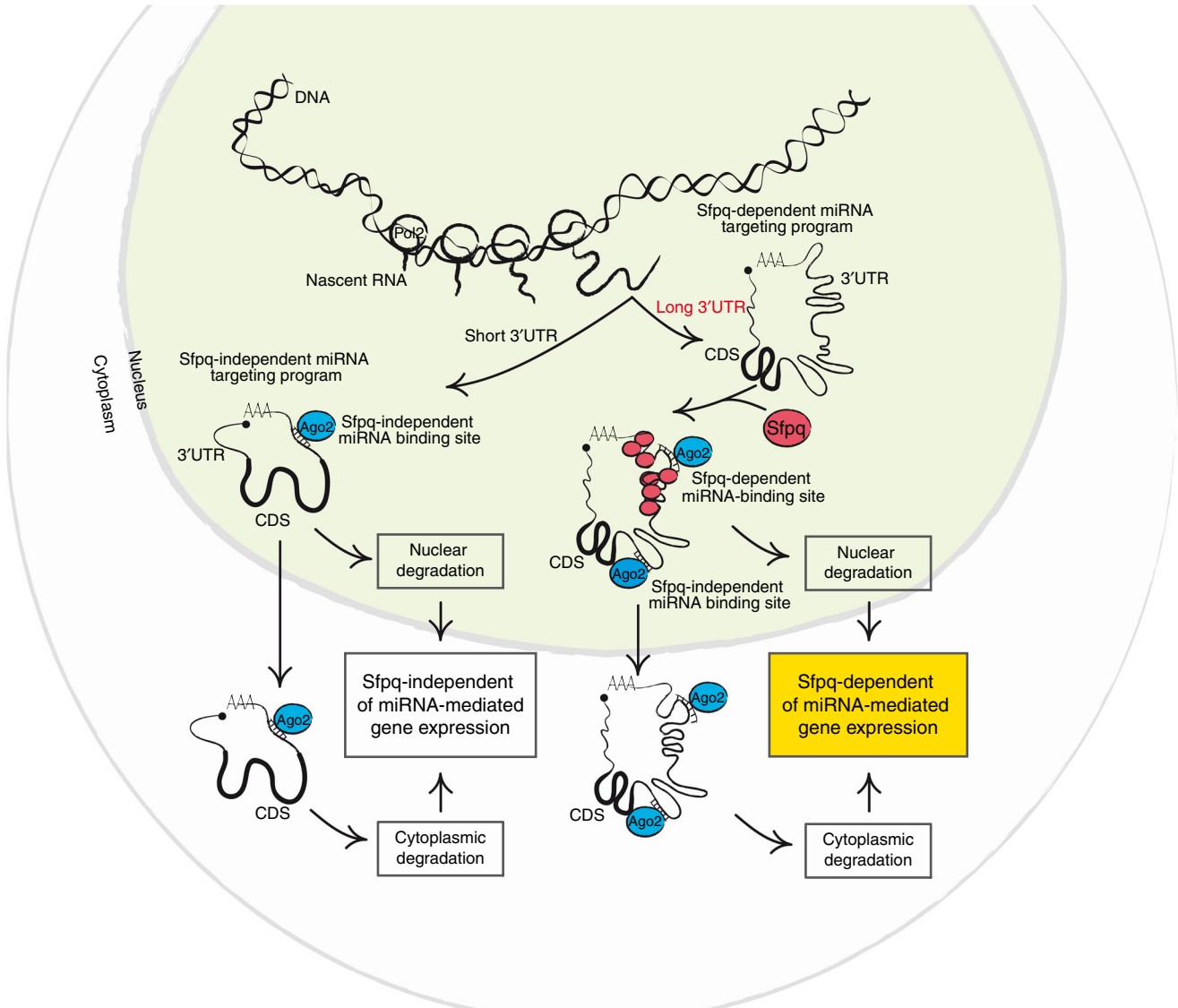

**Fig. 7** A model for the Sfpq-dependent control of miRNA targeting

miRNA targeting, we found that Sfpq targets a sizeable subset of long nuclear 3′UTRs to promote miRNA targeting.

Previous studies have reported the presence of endogenous miRNA pathway components in the nucleus[6, 43, 44]. According to these reports, miRNAs are loaded into miRISC in the cytoplasm and then imported into the nucleus by Importin 8. Indeed, although miRISC and its factors TNRC6 and the CCR4-NOT deadenylase complex are localized in the nucleus, miRISC loading factors, such as Hsp90, are absent[6]. Our data indicate that Sfpq specifically binds to selected long 3′UTRs and forms aggregates that likely modulate the secondary structure of target 3′UTRs to promote optimal positioning/recruitment functions of miRISC to specific binding sites, which leads to miRNA-dependent mRNA degradation. Because Sfpq interacts in an RNA-dependent fashion only with nucleoplasmic miRISC, but not on the chromatin, we concluded that Sfpq only associates with miRNA-target mRNAs post-transcriptionally (Fig. 7). This mechanism is conserved across species and in different cell types (Supplementary Fig. 13a), indicating that our results have uncovered a general strategy by which cells possess a specific pathway to control miRNA targeting in long 3′UTRs via an Sfpq-dependent mechanism. This finding contributes to the diversity of miRNA

modes of action and impacts on post-transcriptional gene expression regulation. Whether the partners of Sfpq, namely, Pspc1, NonO, and the long non-coding RNA Neat1, are part of this mechanism is still an open question.

A plethora of observations accumulated over many years strongly suggest that there is an intimate link between the various steps in the post-transcriptional gene expression pathway, including mRNA export, translation, stability, and localization[45]. Here, we demonstrated that nucleoplasmic Sfpq determines the fate of a sizeable set of mRNAs by acting at the 3′UTR and influencing both nucleoplasmic and cytoplasmic miRNA targeting, including with Lin28A mRNA in stem cells. These data mechanistically reveal for the first time an Sfpq-dependent link between nuclear and cytoplasmic miRNA-dependent silencing (Fig. 7). In conclusion, our findings widen the investigation of post-transcriptional silencing beyond the traditional cytoplasmic point of view to include a nucleoplasmic mechanism.

## Methods

**Proteomic analysis**. The antibody we used to immunoprecipitate Ago2 has been extensively used to immunoprecipitate endogenous Ago2 in several HITS-CLIP analyses, including[46, 47], but it has not been used for MS analysis. IP by IgG and

protein G served as controls, as well as just the anti-Ago2 antibody not incubated with cell lysate. As a quality control for the specificity of the IP, we tested the immunoprecipitated pellet by Northern blotting against let-7a (Supplementary Fig. 1c) and by Western blotting against Ago2 (Supplementary Fig. 1d). The immunopellet from either undigested or totally digested RNA was further treated for MS analysis.

To determine the RNA dependency of the Ago2 interactors identified, we summed the abundance scores for each identified protein from the three replicates in each condition as one single value and used the base 2 logarithm of the ratio of the abundances of the (+)RNase and (−)RNase conditions. As a standard method for RNA-dependent association with the immunoprecipitated protein (Ago2), we normalized the RNA-dependency ratio to set the ratio of Ago2 itself to one based on the assumption that enrichment of the immunoprecipitated protein itself should not be RNA-dependent. According to Klass et al.[48], we set up a cutoff for grouping the proteins as RNA-dependent or RNA-independent interactors by generating a null distribution by modeling RNA-independent associations (red line in Supplementary Fig. 13b). To this goal, we made the following two assumptions: (i) after normalization, any RNA-dependent values more than zero have a true value of zero and the observed variation from zero is due to noise; and (ii) the noise is symmetric about zero. Using these two assumptions, we took the RNA-dependence values higher than zero and built the null distribution symmetric about zero. Thus, we calculated the LOD (log of odds) for each interactor in the RNA-dependent distribution (black line in Supplementary Fig. 13b) vs. the null distribution (red line). Those interactors with a greater likelihood of being in the RNA-dependent distribution (LOD < 0) were considered Ago2 interactors in an RNA-dependent fashion (dashed line in Supplementary Fig. 13b and Supplementary Data 3).

**Mass spectrometry**. For MS, proteins from immunoprecipitated samples were loaded on Bis–tris acrylamide gels (Life Technologies) and migrated as soon as proteins stacked in a single band. Proteins were stained with Imperial Blue (Pierce, Rockford, IL), excised from the gel and digested with high sequencing grade trypsin (Promega, Madison, WI, USA). Then, we performed MS analysis according to Shevchenko et al.[49]. Briefly, gel excisions were washed and destained with 100 mM $(NH_4)HCO_3$, shrunk using 100 mM ammonium bicarbonate in 50% acetonitrile and let them dry at room temperature. After rehydratation using 10 mM DTT in 25 mM ammonium bicarbonate pH 8.0 for 45 min at 56 °C the solution was replaced by 55 mM iodoacetamide in 25 mM ammonium bicarbonate pH 8.0 for 30 min at room temperature in the dark. Excised gels were then washed twice in 25 mM ammonium bicarbonate and shrunk for 5 min in 25 mM ammonium bicarbonate in 50% acetonitrile. They were then reswollen in 25 mM ammonium bicarbonate pH 8.0 supplemented with 12.5 ng μl$^{-1}$ trypsin (Promega) for 1 h at 4 °C and incubated overnight at 37 °C. Finally, digested peptides were harvested by two extractions: the first in 5% formic acid and the second one in 5% formic acid in 60% acetonitrile. Both extracts were pooled and dried.

The resulting samples were injected in duplicate in 0.1% TFA 4% acetonitrile and analyzed by liquid chromatography (LC)–tandem mass spectrometry (MS/MS) in an LTQ-Orbitrap-Velos (Thermo Electron, Bremen, Germany) online with a nanoLC Ultimate 3000 chromatography system (Dionex, Sunnyvale, CA). Peptides were separated on a Dionex Acclaim PepMap RSLC C18 column, according to the manufacturer's instructions. For peptide ionization in the nanospray source, spray voltage was set at 1.4 kV and the capillary temperature at 275 °C. Instrument method for the Orbitrap Velos was set up in data-dependent mode to switch consistently between MS and MS/MS. MS spectra were acquired with the orbitrap in the range of $m/z$ 400–1700 at a FWHM resolution of 30,000 measured at 400 $m/z$. 445.120025 ions were used as lock mass for internal calibration. The 10 abundant precursor ions were selected and collision-induced dissociation fragmentations were performed in the ion trap on the 10 most intense precursor ions measured to have maximum sensitivity and maximum amount of MS/MS data. 500 counts was set as signal threshold for an MS/MS event. Charge state screening was enabled to exclude precursors with 0 and 1 charge states. Dynamic exclusion was enabled with a repeat count of 1, exclusion list size 500 and exclusion duration of 30 s.

**Bioinformatic analysis**. Raw files generated from MS analysis were processed with Proteome Discoverer 1.4 (Thermo Fisher Scientific). This software was used to search data via in-house Mascot server (version 2.4.1; Matrix Science Inc., London, UK) against the mouse subset (16,718 sequences) of the SwissProt database (2015_06). Database search were done using the following setting: a maximum of two trypsin miscleavages allowed, methionine oxidation, N-terminal protein acetylation as variable modifications, and cysteine carbamido-methylation as fixed modification. A peptide mass tolerance of 6 ppm and a fragment mass tolerance of 0.8 Da were used for search analysis. Only peptides with high stringency Mascot score threshold (identity, false discovery rate (FDR) <1%) were selected and used for protein identification (unique peptides).

Relative intensity-based label-free quantifications (abundance scores) were processed using Progenesis LC-MS software (version 4.1; Nonlinear Dynamics, Newcastle, UK), according to the manufacturer's instruction (Nonlinear Dynamics, Newcastle, UK). First, raw LC Orbitrap MS data were imported and LC-MS heatmap of retention time and $m/z$ were generated. Features from these LC-MS were automatically aligned and filtered to retain signals crossing the following

parameters; retention time 10–110 min, $m/z$ 300–1700, charge state 2–6 and 3 or more isotopes. MS-MS spectra were exported into peak list as Mascot generic files (MGF) to allow protein identification using the following inclusion options: the three highest intensity precursors for each feature and precursor intensity over 25%. MGF files were used to search data via in-house Mascot server version 2.4.1 (Matrix Science Inc., London, UK) against the Mouse database subset of the SwissProt database (version 2015_06). Only peptides (unique peptides) adjusted to 1% FDR (identity) and with ion score cut-off of 30 were exported from Mascot results and imported back to Progenesis LC-MS for protein grouping and quantification. Total ion intensity signal from each of the individual peptides generated protein quantification. Any conflicting peptide identifications were removed from the measurements of the quantified proteins. Univariate one-way analysis of variance (ANOVA) was performed within Progenesis LC-MS to calculate the protein $p$-value according to the sum of the normalized abundances across all runs.

We analyzed the enrichment of gene ontology (GO) terms, using the tool GoTermFinder[50], and protein domains by protein families database, using the tool SMART[51] and python in-house scripts (Supplementary Data 3). The majority of Ago2 interactors identified were RNA-binding proteins. Although, our data clearly demonstrated an enrichment of RNA-related terms and domains for both RNA-dependent and RNA-independent interactors of Ago2, the proteins showing RNA-dependency were particularly enriched in post-transcriptional regulation terms and molecular functions, such as mRNA stabilization and mRNA binding (Supplementary Fig. 1a). In addition, RNA-dependent interactors were significantly enriched for known RNA protein-binding domains, including RRM and K Homology domain (Supplementary Fig. 1b), demonstrating the efficacy of the method used here.

**HITS-CLIP method and analysis**. We used a protocol according to Darnell's laboratory[52] and further modified by Wang et al[53]. Briefly, 10 cm diameter dishes of P19 cells were transfected with combinations of mimic let-7a, siSfpq, siCtr, or mimic Ctr for 48 h to select the following conditions: (i) sicontrol (siCtr) and mimic Ctr; (ii) siCtr and mimic let-7a; (iii) siSfpq and mimic Ctr; (iv) siSfpq and mimic let-7a. All four conditions were used to perform Ago2 HITS-CLIP analysis, whereas only the first and the third conditions were used to perform Sfpq HITS-CLIP analysis. Afterward, cells were irradiated once at 400 mJ cm$^{-1}$ with 254 nm UVC light and lysate by scraping them. Lysates were then RNase digested using 10 μg ml$^{-1}$ of RNase I. Crosslinked Ago2 or Sfpq were recovered via overnight IP at 4 °C with 1 μg of either monoclonal anti-mouse Ago2 antibody (Wako Chemicals) or a mixed of two anti-Sfpq antibodies (Ab38148 from Abcam and B92 from Fischer Scientific) complexed with Protein-G Dynabeads. Immunoprecipitates were washed three times with High-Salt Buffer and PNK buffer, end labeled with 25 μCi $^{32}$P gamma-ATP using PNK for 20 min at 37 °C then with 1 mM of cold ATP for 20 min at 37 °C. After three washes with PNK buffer the protein–RNA complexes were eluted from the beads using 1 × NuPage Loading buffer supplemented with 10% of 2-mercaptoethanol at 70 °C for 10 min. Protein–RNA complexes were then resolved on a 10% Bis-Tris Gel and transferred to nitrocellulose. Membranes were exposed onto a cassette at −80 °C for 1–2 h to obtain an autoradiograph film. Subsequently, protein–RNA complexes migrating in the 90–130 kD or 75–110 kD ranges for Ago2 and Sfpq, respectively, were excised (Supplementary Fig. 13c, d). RNA was recovered from the nitrocellulose using Proteinase K treatment followed by phenol–chloroform extraction and ethanol precipitation. Isolated RNA was subjected to small RNA library preparation with the CleanTag™ Ligation Kit for Small RNA Library Prep (Trilink Biotech) using a ¼ dilution of 3′ and 5′ adapters. cDNA products were subjected to PCR amplification for 20 or 22 cycles for Sfpq and Ago2 experiments, respectively, and were then purified on Purelink PCR micro kit columns (Invitrogen). PCR products represented cDNA inserts of 20–50 nt. Libraries were quantified using both bioanalyzer (Agilent) and Qubit (Invitrogen) and were subjected to sequencing by Ion Proton sequencer (Life Technologies).

**Bioinformatic analysis**. Reads were mapped onto the mm10 genomes using STAR[54], then unmapped reads were mapped with bowtie2[55]. Total number of reads mapped are reported in Supplementary Data 5. The reproducibility of the replicates for each condition was assessed by performing principal component analysis using the R package "htSeqTools" (R package version 1.14.0). Afterward, replicates were merged together to assess bioinformatics analysis. Ago2 peaks specific of let-7a-transfected cells condition were found by comparing let-7a-transfected samples with the control using dCLIP program[56]. We filtered out peaks with score less than 7 and have identified 4202 Ago2-differential binding peaks (here called Ago2-let-7a peaks, Supplementary Data 5). To identify let-7a canonical-binding sites, we screened the identified Ago2-let-7a peaks sequences in the 3′ UTR looking for the reverse complement of the seed sequence (UACCUC). Identical approach was used to identify the canonical-binding sites for the 21 most endogenously expressed miRNAs (Supplementary Data 5). To map Sfpq-binding sites we used dCLIP program by comparing Sfpq HITS-CLIP in siSfpq-transfected P19 cells with siCtr-transfected cells. We only selected peaks with a score greater than 7.

To investigate the potential role of Sfpq to promote let-7a targeting, we checked whether the identified Ago2-let-7a peaks were reduced in absence of Sfpq. We first selected the Ago2-let-7a peaks enriched in let-7a-transfected P19 cells condition

compared to let-7a-siSfpq-transfected P19 cells by using dCLIP program. Second, we selected Ago2-let-7a peaks that do not show any change when comparing siSfpq-transfected P19 cells with the siCtr condition by using dCLIP program. We reasoned that in the first filter we identify all Sfpq-dependent Ago2-let-7a peaks, whereas in the second filter we double check the real let-7a dependency of the identified peaks. To merge the information from the two filters, we measured the overlap between the full set of 4202 Ago2-let-7a peaks with the peaks coming from these two filters by calculating Jaccard index (minimal threshold index of 0.10 for overlapping peaks). At the end, we found that 3381 Ago2-let-7a peaks overlap with the peaks coming from these two filters, indicating their dependency on Sfpq.

To assess the Sfpq dependency of the endogenous miRNA-binding sites, we selected Ago2 peaks that overlap (Jaccard index > 0.10) between the full set of 2065 Ago2-miRNA peaks identified by HITS-CLIP analysis from P19 cells[34] and the Ago2-miRNA peaks enriched in siCtr-transfected P19 cells over the siSfpq-transfected P19 cells condition identified by dCLIP analysis. We found that 245 Ago2-miRNA peaks overlap with the peaks coming from such a filter, indicating their dependency on Sfpq.

Ago2-let-7a peaks recruited in absence of Sfpq were identified by comparing siSfpq-let-7a-transfected P19 cells to let-7a-transfected P19 cells with dCLIP analysis and selecting peaks with a score greater than 7. For bindings sites of endogenously expressed miRNAs, we analyzed Ago2-miRNA peaks in siSfpq-transfected P19 cells using pyicoclip[57] with the same protocol as described in Bottini et al[34]. In Supplementary Data 7 is shown the recruited Ago2-let-7a or Ago2-miRNA peaks in siSfpq-transfected P19 cells.

Spatial correlation of the Sfpq-dependent recruited Ago2 peaks (for both let-7a and endogenous miRNAs) compared to the decreased ones was performed with the R package GenometriCorr[58]. Briefly, we found that the two sets of Sfpq-dependent recruited and reduced Ago2 peaks are spatially correlated across the genome, calculated the significance as a deviation from a non-uniform distribution of one set of peaks over to the other one.

All analyses were performed with custom python scripts where not specified elsewhere.

**User-friendly website for Ago2 and Sfpq CLIP data sets**. We created a database in MySql language with a web interface developed in php language to easily access to the entire mapping of Sfpq, Ago2-let-7a, or Ago2-miRNA peaks identified by HITS-CLIP analysis. The database can be interrogated by the following four different keys: (1) "gene identifier", which provides all peaks on the requested gene; (2) "genomic category", which provides all peaks that map in specific genomic category, including 3′UTR, 5′UTR, CDS, introns, TTS, promoters, intergenic, and ncRNA; (3) "miRNA name", to access to all peaks bearing the signature of let-7a or the 21 most expressed miRNAs in P19 cells (Supplementary Data 5); and (4) "peak category", which provides all peaks that belong to a specific category as assessed by this study, including Sfpq peaks, Ago2 peaks, Sfpq-dependent decreased Ago2 peaks localized within 500 nt from Sfpq peaks, Sfpq-dependent decreased Ago2 peaks localized between 500 and 7000 nt from Sfpq peaks, and Sfpq-dependent decreased Ago2 peaks localized more than 7000 nt from Sfpq peaks. One or more keys can be combined to explore the data from different points of view. The database is freely accessible at http://trabucchilab.unice.fr/SITO/index.php#.

**Search for Sfpq-binding motifs and clusterization**. To identify the putative binding motif(s) of Sfpq that would play a role in miRNA targeting in the 3′UTR, we divided the data set of Sfpq peaks from the HITS-CLIP analysis in the following three subsets: (i) all Sfpq peaks; (ii) Sfpq peaks located in the 3′UTR; and (iii) Sfpq peaks located in the 3′UTR within a distance of 500 nt from the Sfpq-dependent reduced Ago2-let-7a peaks. Then, we launched de novo motif discovery with a length between 4 and 6 nt. We used two motif finding programs: Homer[59] and DREME[60]. We observed that all these analyses reported significantly enriched motifs with a core composed by UGU. Thus, we performed motif clusterization with the program "Gimmemotifs"[41] (Supplementary Fig. 10a) and found two consensus motifs of 4 and 5-nt long, respectively (Fig. 6d). Each motif corresponds to many different words that belong to the motif with different probabilities. Hence, to set up a threshold on the number of words to look for, we calculated the occurrences of the two motifs in the sequences of the three Sfpq peak subsets previously selected. We used the tool FIMO[61] using different p-value cutoffs associated to the motif likelihood established by the program. For cutoffs higher than 0.01 for motif#1 and 0.005 for motif#2, we found many occurrences superimposed. Thus, to avoid superimposition, we established the aforementioned cutoffs, which correspond to two words for the motif#1, namely, CUGU and CUGA, and four words for motif#2, namely, CUGUA, AUGUA, UUGUA, and GUGUA.

**RNA-IP and RNA sequencing**. For RNA-IP, cells lysates were immunoprecipitated with Protein G-Dynabeads (Invitrogen)-coupled antibodies at 4 °C overnight[2]. When required, cell lysates were incubated at room temperature with RNase A (10 μg ml$^{-1}$, Ambion) for 30 min for partial RNA digestion before the IP. When indicated, cell lysates were also incubated at room temperature with 100 nM recombinant human Sfpq wild-type or Sfpq-214–598 quadruple mutant (L535A, L539A, L546A, and M549A) for 30 min before RNA digestion and IP. Total RNA

was prepared using Trizol (Invitrogen) and RNA columns (Qiagen), and analyzed by either small RNA sequencing or quantitative RT-PCR.

For small RNA sequencing 45 ng of RNA from each IP were subjected to small RNA library preparation with the CleanTag™ Ligation Kit for Small RNA Library Prep, using a ¼ dilution of the 3′ and 5′ adapters and performing 15 cycles of PCR. PCR products were purified on Purelink PCR micro kit columns (Invitrogen). Final libraries were quantified with bioanalyzer (Agilent) and Qubit (Invitrogen), and subjected to sequencing by Solid sequencer (Life Technologies). Sequenced fragments were mapped with the LifeScope 2.5.1 pipeline (Life Technologies) and annotated according to the Ensembl non-coding RNA database.

*Bioinformatics analysis*: First, we checked the quality of the replicates using the PCA, then we used the webserver omiRas[62] to map the reads on the mouse genome, to normalize the data, and to compare each RNA-IP sample to the input. To avoid detecting differences at the level of transcriptional noise, we required a minimum of 20 reads for each small RNA identified. In order to specifically select small RNAs enriched in RNA-IP samples respect to the input, we employed Gaussian mixture modeling of the small RNA population distribution (Supplementary Fig. 5a) and calculated the LOD to filter out the background distribution (LOD > 0) from the input (Supplementary Data 4).

**RNA-sequencing experiments and analysis**. To check whether Sfpq may control miRNA targeting by alternative splicing or polyadenylation sites, we performed a RNA-sequencing analysis from siSfpq-transfected P19 cells and control using the Truseq Stranded mRNA kit (Illumina) from 1 μg total RNA and performing 15 PCR cycles. The final library size was centered on 290 bp. Paired-end sequencing was performed on a NextSeq 500 sequencer (Illumina) on a Mid 150 flowchip. The quality of replicates was assessed using the R package "htSeqTools". Normalization of RNA-seq data was performed with the DEseq package available from Bioconductor (http://www.bioconductor.org). To identify differential splicing events between samples, we used the software SpliceTrap[63] (Supplementary Data 8). To assess whether Sfpq interferes with alternative polyadenylation sites and the length of the 3′UTRs we used the software DaPars[64]. Our analyses did not find any differential splicing or shortening-lengthening events regulated by Sfpq in those 3′ UTRs on which Sfpq promotes miRNA targeting (Supplementary Fig. 12a).

**Additional bioinformatics and statistical analyses**. GO-term analysis was done with the webtool "GoTermFinder"[50] and the pictures were generated using the tool REVIGO[65] and custom R script. miRNA seed searching in peak sequences was performed with a custom python script that takes as input the peak sequences and the miRNA sequences, then calculates the reverse complement of the seed region using the library "Biopython"[66]. Venn diagrams were calculated with the webtool "Venny" (http://bioinfogp.cnb.csic.es/tools/venny/index.html). All statistical analyses were performed with the statistical software R, all other analyses were performed with custom python scripts where not specified otherwise.

**Microarray analysis**. For mRNA profiling analysis three replicates of P19 cells were transfected with siSfpq, mimic let-7a, siSfpq + mimic let-7a, or control (Supplementary Data 6). 48 h after transfection, total RNA was isolated by using RNAeasy kit (Qiagen). RNA samples were labeled with Cy3 dye using the low RNA input QuickAmp kit (Agilent) as recommended by the supplier. 400 ng of labeled cRNA sample was hybridized on 8 × 60 K high-density SurePrint G3 gene mouse GE 8 × 60 K Agilent microarrays. Assessment of the quality of replicates was performed using the Bioconductor package "ArrayQualityMetrics". Normalization of microarray data was performed with the Limma package available from Bioconductor (http://www.bioconductor.org) using the quantile methods.

**Cellular fractionation**. Cellular fractionation of cytoplasm, nucleoplasm, and chromatin were performed as previously described (ref.[67,68]), with a modification for the chromatin solubilization according to a previous protocol. Briefly, genomic DNA pellet was resuspended for treatment with Turbo DNase I in the DNase I digestion buffer (50 mM Tris pH 7.5, 0.5% Nonidet-P40, 0.1% sodium lauroyl sarcosine, 1 × Complete protease inhibitors) at 37 °C for 45 min with intermittent vortexing. The genomic DNA was further solubilized by adding 1% SDS, 0.3 M lithium chloride, 25 mM EDTA, and 25 mM EGTA and incubated at 37 °C for 15 min.

For IP experiments the salinity of 300g of protein lysate from cytoplasmic, nucleoplasmic, orgenomic DNA solutions was adjusted to 150 mM of NaCl.

Cellular fractionation for Fig. 5D was performed according to the protocol of Core et al. (ref.[68]). For Supplementary Fig. 4c, fractionation followed the method of Gagnon et al. (ref.[67]), with the chromatin solubilization step adapted from Minajigi et al. (ref.[69]). Specifically, the chromatin pellet was resuspended in 50 mM Tris-HCl (pH 7.5), 0.5% NP-40, and 0.1% sarkosyl, then sonicated, treated with DNase I, and supplemented with LiCl (final 300 mM) and 1% sarkosyl. Protease inhibitor cocktail (Roche) and RNaseOut (for RNA analyses) were included in all buffers.

**Reagents and antibodies**. Glutathione-agarose, Actinomycin D, mouse IgG, rabbit IgG, anti-Flag, and anti-GST antibodies were purchased from Sigma. Knockdown experiments were performed with Smartpool siRNAs (Thermo Fisher Scientific), namely, mouse Sfpq (L-044760-01 and E-044760-00), human Sfpq (E-006455-00 and L-006455-00), mouse Pspc1 (L-49216-01), and mouse NonO (L-048587-00). Each catalog number of Smartpool siRNAs corresponds to a set of four different siRNAs. Two different Smartpool siRNA catalog numbers that target the same gene are composed by different siRNAs.

Protein-G Dynabeads were purchased from Thermo Fisher Scientific. PNK was purchased from New England Biolabs. Mouse monoclonal anti-Ago2 antibody (2D4) for HITS-CLIP, RNA-IP, and co-IP was purchased from Wako Chemicals. Rabbit anti-Sfpq (Ab38148), goat NonO (ab50411) antibodies were from Abcam.

Mouse monoclonal anti-c-Myc (9E10), mouse anti-BrdU, mouse anti-Tubulin, goat anti-GST, goat anti-actin (I-19), Rabbit anti-Pspc1 (sc-84577), and mouse anti-Pspc1 (sc-374387) antibodies were from Santa Cruz. Mouse anti-Sfpq antibody (B92) was purchased from Fischer Scientific. Mouse anti-Ago2 antibody for IP in human cells and immunoblotting was purchased from Sigma (WH0027161M1). Rabbit anti-Ago2 antibody for immunostaining and immunoblotting was purchased from Cell Signaling (C34C6). The immunofluorescence was revealed by using fluorescent secondary antibodies: Alexa Fluor 488-conjugated anti-mouse IgG (H + L) antibody (Cell Signaling), Alexa Fluor 594-conjugated anti-rabbit IgG (H + L) antibody (Molecular Probes). Slides were coverslipped in Vectashield Mounting Medium with DAPI (Vector Laboratories). Primary antibodies were diluted to 1 µg ml$^{-1}$ or 0.1 µg ml$^{-1}$ for immunoblotting or immunofluorescence, respectively. Secondary antibodies were diluted, according to the manufacturer's instructions.

**Cell transfection and immunoblotting**. RAW 264.7, P19, NTERA-2, HeLa, and HEK293T cells were purchased from ATCC. No mycoplasma contamination was assessed. Cells were transiently transfected for 48 h with Lipofectamine RNAiMax or 2000 (Invitrogen), according to the manufacturer's instructions. siRNAs and mimic miRNAs were transfected at the final concentration of 24 nM. Plasmids were transfected at the final concentration recommended by the manufacturer. When required, cell lysates were incubated at room temperature with RNase A (10 mg ml$^{-1}$, Ambion) for 30 min for full RNA digestion. Three-hundred micrograms of proteins were immunoprecipitated with Protein G-Dynabeads (Invitrogen)-coupled antibodies for 16 h at 4 °C with rotation. Immunoprecipitates were washed four times with the lysis buffer and resuspended in SDS protein loading buffer. Proteins were subjected to SDS-PAGE, electroblotted onto PVDF membranes, and probed with the indicated antibodies. Western blot analyses were performed with 30 µg of protein lysate where not specified otherwise.

**Immunofluorescence**. For immunofluorescence, cells were fixed in 10% neutral formalin and incubated overnight with the primary antibodies[70]: anti-Sfpq, anti-Pspc1, or anti-Ago2 antibodies. Slides were coverslipped in Vectashield Mounting Medium with Dapi (Invitrogen). As control for immunofluorescence, slides were incubated with only the Alexa Fluor 488-conjugated anti-mouse or Alexa Fluor 594-conjugated anti-rabbit antibodies but not with primary antibodies. No signal was detected in these conditions. The results were analyzed on a Leica DM5500B microscope with a HAMAMATSU camera ORGA-ER.

**Luciferase reporter assay**. HEK293T cells (80% confluence in 96-well plates) were transfected with Lipofectamine 2000®, according to the manufacturer's instructions. Luciferase reporter assay was performed according to the manufacturer's instructions.

**RNA EMSA**. $^{32}$P-labeled RNAs were synthesized and used as substrates for EMSA[71]. Briefly, $^{32}$P-labeled RNAs were incubated with recombinant protein in the binding buffer (75 mM Hepes 7.9, 150 mM NaCl, 75 mM KCl, 0.1 mM EDTA, 5 mM DTT, 5 mM PMSF, and 15% glycerol) for 15 min at room temperature. The $^{32}$P-labeled RNA-recombinant protein complex was resolved in a 10% acrylamide native gel. Finally, gel was dried and exposed at different time onto a film.

**Northern blot**. Total RNA was isolated from cells using Trizol (Invitrogen), resolved on 10% polyacrylamide-urea gels, and electroblotted onto HyBond N+ membranes. Membranes were hybridized overnight with radiolabeled DNA antisense oligonucleotides to miRNAs in ExpressHyb solution (Clontech). After hybridization, membranes were washed three times with 2× SSC and 0.05% SDS, twice with 0.1× SSC and 0.1% SDS, exposed overnight onto a film. The same blot was hybridized (upon stripping in boiling 0.1% SDS) with up to three distinct oligonucleotide probes (Supplementary Table 1).

**mRNA and miRNA quantitative expression analysis**. RNA expression by quantitative RT-PCR was performed by using standard procedures. Briefly, total RNA was isolated from cells using miRNeasy kit (Qiagen) and cDNA was synthesized with a random hexanucleotides using Superscript III or n-code kit for miRNAs (Life Technologies). Quantitative RT-PCRs using Sybr Green were performed on a StepONE system (Applied Biosystem). Expression was considered undetectable with Ct value ≥ 40. The target expression value was normalized with a couple of reference genes: either U6–U2 snRNAs or actin–beta-microglobulin. The relative expression level was further normalized by the $2^{-\Delta\Delta Ct}$ method when indicated. The Student's t-test or the one-way ANOVA followed by Tukey's post hoc test was performed to assess statistical significance. The primer sequences are detailed in Supplementary Table 1.

**Nuclear run-on**. We performed NRO followed by RT-qPCR[72]. Briefly, nuclei were prepared as described in "Cellular fractionation" method. To produce nascent RNA, nuclei were incubated with the run-on buffer (5× buffer: 0.5 M Tris pH 8, 1 M Mg$_2$Cl$_2$, 1.5 KCl, 200 mM DTT, 100 mM rNTPs, and 2 mM BrdUTP) for 30 min at 30 °C. Trizol-extracted RNA was immunoprecipitated with anti-BrdU

antibody overnight and eluted. RT-qPCR was performed from input and immunoprecipitated samples. Five µl of 1.5 pg µl$^{-1}$ in vitro transcribed 300 nt spike-in BrU-GFP (positive control) and fire-fly luciferase (FF—negative control) were added to 100 µl of extracted NRO-RNAs (final concentration 0.15 pg µL$^{-1}$). Nascent UTP-actin or BrU-actin B was used as internal control for NRO experiments (Supplementary Fig. 9c, right panel).

**Plasmids and recombinant proteins**. pMIR-REPORT plasmid containing the wild-type sequence of mouse Lin28a 3′UTR downstream the fire-fly luciferase CDS for gene reporter assay was kindly gifted by Dr. Baltimore. pet15b plasmid containing 6×His-tagged full-length human Sfpq or the Sfpq-214–598 quadruple mutant (L535A, L539A, L546A, and M549A) were kindly provided by Drs. Lynch and Bond, respectively. Sequence encoding only for the two RRM domains of human Sfpq was cloned in-house into pGEX-4T plasmid (GE Healthcare Life Sciences). Production of recombinant proteins were performed by the Protein Expression Facility at the University of Aix-Marseille. The purity of the recombinants was tested with protein gel stained by Imperial or Coomassie blue dye or by Sfpq immunoblotting. Recombinant Ago2 was purchased from Sino Biological Inc. Luciferase reporter gene plasmid containing 6xlet-7a-binding sites was kindly gifted by Dr. J.G. Belasco. HA-Sfpq was generously provided by Dr. S. Saccani. MYC-Ago2, Flag-Ago2, and Flag-Ago1 were provided by Addgene. Oligonucleotides for cloning are listed in Supplementary Table 1.

**Atomic force microscope**. The in vitro transcribed Lin28A 3′UTR was heated at 95 °C for 1 min and immediately cooled in ice. We then added the binding buffer (120 mM NaCl, 24 mM Tris-HCl pH 7.5, 4 mM MgCl$_2$) containing the recombinant Sfpq proteins (wild-type or Sfpq-214–598 quadruple mutant (L535A, L539A, L546A, and M549A)) or BSA to a final protein-RNA concentration ratio corresponding to 121 nM/13 nM, and incubated at 4 °C for 30 min. Ten µl of this solution was applied to the untreated muscovite mica surface and incubated for 10 min. After the incubation, we washed the surface with 1 ml of distilled water to wash away the unbound RNA and dried under a gentle flux of nitrogen gas. In our experimental condition, the mica surface is not enough positively charged to allow the absorption of unbound RNA, which is removed from the surface in the washing step. Thus, RNA is only visible on the mica surface upon interaction with protein.

Imaging was performed using the Multimode8 AFM connected to the Nanoscope V controller (Bruker Nano Surfaces, Santa Barbara, CA, USA) equipped with E-scanner, operating in PeakForce mode in air at room temperature using ScanAsyst-air probes (Bruker Nano Surfaces, Santa Barbara, CA, USA). Images of 1 µm$^2$ corresponding to 512 × 512 pixels were captured with a scanning rate of 1 Hz using a PeakForce frequency of 2 kHz, a PeakForce setpoint of 0.03 V, and a Z limit of 700 nm. Raw AFM images were processed only for background slope removal (flattening).

**Data availability**. All sequencing and microarray for mRNA profiling were submitted to the GEO database under accession number series GSE89033.

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

## Acknowledgements

We are indebted to Drs D. Baltimore, J.G. Belasco, K.W. Lynch, C.S. Bond, and S. Saccani for reagents and to Mr. G. Bottini for the user-friendly interface website. We thank Drs D. Alcor for imaging facility and D. Byrne for protein expression facility. This work has been supported by the AVENIR program of INSERM and Marie Curie CIG to M.T., ANR through the "Investments for the Future" #ANR-11-LABX-0028-01 (LABEX SIGNALIFE) to M.T. and P.B., ITMO cancer of Avesian within the framework Plan Cancer 2009–2013 (ProstaMir P029393) to M.T., P.B., and M.B. N.T. was supported by INSERM-PACA region fellowship, M.T. by FRM (grant #DEQ20140329551), P.B. by FRM (grant #DEQ20130326464), and E.R. by FRM (ING20140129224). Marseille Proteomics (IBiSA) is supported by Institut Paoli-Calmettes and Canceropôle PACA. PICMI, the IRCAN's Imaging facility is supported by ARC, IBiSA and Conseil General 06 de la Région PACA. Sequencing was performed by the UCA GenomiX platform, supported by ANR (ANR-11-LABX-0028-01, ANR-10-INBS-09-03, and ANR-10-INBS-09-02) and Canceropole PACA.

## Author contributions

M.T. conceived and coordinated the study, and designed the experiments. N.H.-T. performed the majority of the experiments. R.M., E.R. and M.T. performed the experiments. L.-E.Z. and P.B. performed high-throughput sequencing. S.A. performed the mass-spectrometry experiments. S.P. performed the AFM experiments. S.B., E.R. and M. T. analyzed the data. V.G., M.B. and C.M. participated in data analysis. S.B. performed the bioinformatics analyses. M.T. wrote the manuscript. All authors read and approved the final manuscript.

## Additional information

**Competing interests:** The authors declare no competing financial interests.

