## [Peer Review File · Nature Communications]

Reviewers' Comments:

Reviewer #1 (Remarks to the Author)

This manuscript analyzes the role of Sfpq in modulation of miRNA mediated silencing, using a number of approaches including IP/MS, HITS-CLIP, reporter assays, EMSA, AFM. The authors show that: (1) Ago interacts with Sfpq via RNA (Ago IPs; Fig. 1-3); (2) Sfpq binds to a subset of Ago targets (HITS-CLIP; Fig. 4); (3) KD of Sfpq leads to upregulation of mRNA bound by let-7 and Sfpq (Fig. 5). (4) Sfpq preferentially binds longer 3'UTRs; the binding motif contains UGU; the protein can form aggregates. The question addressed is interesting and important, but data presentation and interpretation needs improvement, especially the mechanistic part.

Specific comments:

Fig. 3A-B: I understand that the authors detect Ago and component of Sfpq complex in the same cellular fraction (nucleoplasm). I do not think it is enough to talk about their co-localization though, which implies higher resolution technique than the ones applied by authors. For example, double staining IF.

Fig. 4D: There is an error in the pie charts. For example 3'UTR (35%) slice is smaller than intron (27%) in the third pie chart.

Fig. 5A and S7A: aims to show that KD of Sfpq leads to upregulation of let-7 targets bound by Sfpq. However none of the figures include another group of transcripts that would allow us to judge about the specificity of observed effect. Do other let-7 targets, not bound by Sfpq according to HITS-CLIP, stay unaffected? The authors should choose an appropriate way to estimate the specificity of effects of Sfpq KD. For example, plot HITS-CLIP data (peaks/input or other normalizer for Ago and Sfpq) against RNAseq (Sfpq KD/control).

Fig. 6: It is expected that longer 3'UTR would have statistically higher chance to bind any RBP than short 3'UTRs. How were the data in panel C normalized to take that into account? I am not clear about the mechanistic interpretation of presented data. The authors suggest that Sfpq aggregation facilitates let-7 recruitment, but do not provide the evidence that aggregation is required or clear interpretation how exactly Sfpq aggregation would facilitate let-7 recruitment. For example, what is different between "controlling close" and "not controlling close" Sfpq sites? ("24% and 8.2% of Sfpq-independent Ago2/let-7a peaks and endogenous Ago2/miRNA peaks, respectively, are located in the 3'UTR at close distance to Sfpq peaks (< 500 nt), suggesting that only a subset of miRNA binding sites is directly controlled by Sfpq"). For example, could Sfpq affect 3'UTR folding and affect accessibility of miRNA binding sites? The authors should make a clear distinction between cause & consequence and correlative evidences and discuss possible mechanisms of Sfpq function in discussion.

Reviewer #2 (Remarks to the Author)

Bottini et al., explored Ago2-interacting factors in nucleoplasm and cytoplasm. Then the authors identified novel interacting proteins including Sfpq, NonO and Pspc1, and validated the RNA-dependent interactions between Ago2 and Sfpq/NonO/Pspc1 by biochemical experiments. In addition, by Sfpq knockdown experiments, it is shown that Sfpq somehow influences the recruitment of the miRISC to a subset of their binding sites. Also, by using HITS-CLIP and

bioinformatics approaches, the authors try to show their mechanism of action. Overall, the biochemical experiments are clear, but the data related to the molecular mechanism how Sfpq modulates miRNA targeting is not convincing. In addition, some experiments lack essential controls. Furthermore, there are many grammatical errors, typos and unusual term usages, and so I recommend the authors to consider consulting a language editing service. Please refer to specific points listed below.

Major points

1. In the Sfpq HITS-CLIP experiment, this seems to lack important controls though this experiment is one of the major experiments in this manuscript. In Supplemental Methods (page 3), it is described that protein-RNA complexes 75-90 kD range for Sfpq were excised. But, in the corresponding Supplemental Figure 11C, the dashed boxes do not indicate the 75-90 kD, instead around 60-100kD. Which data is correct? It is essential in the HITS-CLIP experiment where are excised from the gel. Related to this point, in Supplemental Figure 11C, the authors show the data using Sfpq knockdown as a control, but it is ambiguous which bands or parts correspond to Sfpq. So, I think that the authors need to show several controls. The first is western blot data to show that the excised protein-RNA complexes contain Sfpq. The second is the data of the Sfpq-IP from cells without UV crosslink to compare with UV cross-linked samples, which is frequently used to show the specificity of the protein-RNA interactions. It is really important to represent that this Sfpq HITS-CLIP experiment was precisely performed.
2. In the protein-IP experiments, it is shown that the interaction between Ago2 and Sfpq is RNA-dependent. But the authors do not address what RNAs mediate this RNA-dependent interactions. The authors performed Ago2 and Sfpq HITS-CLIP, and thus would be able to examine the overlap of their target RNAs, which might mediate the RNA-dependent interaction between Ago2 and Sfpq.
3. The authors define less than 500 nucleotides as "close" in Figure 4. But 500 nucleotides are not so close enough to convince direct functional connections between Ago2 and Sfpq.
4. In Figure 4F and 4G, the authors used recombinant SFPQ in the rescue experiments. But this lacks important negative controls such as unrelated proteins or SFPQ mutants. Without these negative controls, it is possible that the addition of proteins itself increases the targeting of let-7a to the targets.
5. In Figure 6, the authors claim that Sfpq aggregates in a sequence specific fashion onto long 3'UTR promote miRNA targeting. However, no evidence is shown for such a specific interaction of the Sfpq aggregates onto Lin28A 3'UTR and the aggregates of Sfpq actually facilitate the miRNA targeting. To show these points, I think it essential to take precise negative controls and perform new experiments.

Minor points

1. Many of the statistics are not correct. For example, the student t-test is not applicable to multiple samples. Please use appropriate statistics.
2. In page 6, the authors describe "Interestingly, these proteins are known to associate with the long non-coding RNA Neat1 to form the paraspeckle, a nucleoplasmic compartment of approximately 0.2-1 um in size of not yet well understood function". However, if you search NEAT1/Paraspeckle in PubMed, you can find many papers showing their functions in molecular and organismal levels. They should be properly cited.
3. In many Figures, the labels are redundant. For example, in Figure 2 "5% input", "IP-Sfpq", "IP-Pspc1" and "IP-NonO" appear many times. I think it better to simplify the figures by reducing these redundancies. It would make the figures easily readable.
4. In Figure 2F and Supplemental Figure 2D, the authors
5. In Figure 3C, "5% Input Cytoplasm" should be corrected to "Cytoplasm"
6. In Supplemental Figure 2, I think it better that the labels such "Tagged Myc-Ago2" are replaced to "MYC-tagged Ago2" or just "MYC-Ago2".
7. In Figure 7, RNA polymerase II does not open long range of genomic DNA region. Please illustrate it more precisely.
8. RNase should be corrected to RNase.

9. The term "Hek-293T" seems to be unusual. It should be corrected to HEK293T or another prevalent name.
10. In page 7 "(Supplemental Table 4),", ", " is extra.
11. In page 11, "Interestingly, Sfpq peaks in the 3'UTR are longer..." is strange. This should be corrected.
12. In page 12, "2 RRM domains bind 4 nt" is incorrect. According to reference 33, one RRM domain binds 4 nucleotides, and so 2 RRM domains recognize 8 nucleotides.
13. In page 12, "the motifs 4/6 nt in length" means "the motifs 4-6 nt in length"? Please specify it.
14. In Supplemental Methods (page 1), "NH₄HCO₃" should be corrected to "NH₄HCO₃".
15. In Supplemental Methods (page 5), what is "User friendly interface website" for? Please specify this point.
16. In Supplemental Methods (page 8), "Gluthation-agarose" should be corrected to "Glutathione-agarose"

Reviewer #3 (Remarks to the Author)

This is a solid piece of work that suggests a new and exciting example of post-transcriptional gene regulation. Overall, the writing is good but not great and could use considerable editing and smoothing of the english and grammar. The manuscript tends to get very jumbled and may be trying to show too much simultaneously. Overall however, I am comfortable with the work and the conclusions and felt the data convincing, just not presented as well as possible.

Suggestions:

1. The term "dismissed" and "dismissal" is used throughout (i.e. the dismissed Ago2 peaks upon Sfpq knockdown) in a way that seems odd. I am assuming the intended meaning is decreased or reduced but it is not clear. I also recommend that the term "altogether" be replaced with "together."
2. Like wise the authors suggest a role for Sfpq in promoting "nuclear imprinting of mRNA." I feel that this is a misuse of the term "imprinting."
3. Many of the conclusions from data are over interpreted with terms like "consistently, dramatically, major and significantly" not being warranted. The discussion, however is more appropriately tempered.
4. Not sure what "we deeply analyzed..." means and figure 6A-C are over interpreted.
5. The annotation for significance and which bars are being referenced is unclear for most of the data represented as bar graphs (i.e. 4F and 5B).
6. Fig 4A is not represented well.
7. The use of BSA and GST as control proteins for binding assays are not ideal. An alternative RNA-binding protein would be more appropriate.
8. The AFM data is difficult to interpret without appropriate controls. This is not a data that will be familiar to many and It is impossible to know with any certainty that we are really looking at Lin28A 3' UTR and Sfpq aggregates. Control molecules or reference material would be helpful, especially since the Supp AFM data does not have a needed size bar. The overall size of the depicted components in Fig 6F seems questionable given that ribosomes for example are 30nm.

RESPONSE TO REVIEWERS:

Reviewer # 1

We were pleased that the Reviewer found the work of interest. We thank the Reviewer for the questions provided for additional experiments and analyses. This allowed us to better define the relevance of our findings. We hope that the new data/analyses as well as interpretation of the findings have sufficiently addressed the questions raised by the Reviewer and improved the manuscript to the level appropriate for publication.

Specific detailed answers follow:

Fig. 3A-B: I understand that the authors detect Ago and component of Sfpq complex in the same cellular fraction (nucleoplasm). I do not think it is enough to talk about their co-localization though, which implies higher resolution technique than the ones applied by authors. For example, double staining IF.

We have now included in the new Figure 3B of the revised manuscript a co-immunofluorescence technique showing co-localization of Sfpq and Pspc1 with Ago2 in RAW 264.7 cells. These experiments indicate that both Sfpq and Pspc1 co-localize with nuclear Ago2. We have now inserted these experiments in the "Results" section, page 7 lines 12-15 of the revised manuscript.

Fig. 4D: There is an error in the pie charts. For example 3'UTR (35%) slice is smaller than intron (27%) in the third pie chart.

We thank the Reviewer to have spotted this error. Indeed, the pie charts were right, but we wrongly reported the numbers and the percentages of the pie chart entitled "More than 7,000 nt from Sfpq peaks". In the new Figure 4D of the revised manuscript, we have now corrected this error.

Fig. 5A and S7A: aims to show that KD of Sfpq leads to upregulation of let-7 targets bound by Sfpq. However none of the figures include another group of transcripts that would allow us to judge about the specificity of observed effect. Do other let-7 targets, not bound by Sfpq according to HITS-CLIP, stay unaffected? The authors should choose an appropriate way to estimate the specificity of effects of Sfpq KD. For example, plot HITS-CLIP data (peaks/input or other normalizer for Ago and Sfpq) against RNAseq (Sfpq KD/control).

As suggested by the Reviewer, we now show, in the new Supplementary Figure 7B of the revised manuscript, the expression profile of let-7a targets virtually not bound by Sfpq (we considered transcripts with Ago2/let-7a peaks in the 3'UTR having an Sfpq peak with a distance of more than 7,000 nt). We have used cumulative distributions to plot the expression levels of 54 let-7a targets (both canonical and non-canonical binding sites) upon let-7a transfection (red line) against let-7a and siSfpq transfection (orange line) in P19 cells, to estimate the specificity of Sfpq knockdown effects. As shown in this plot, no statistically significant difference is reported between the two curves, demonstrating that the absence of Sfpq does not affect the global expression levels of this group of Sfpq-independent let-7a targets (Wilcoxon test, p value = 0.9239). Because only two canonical let-7a binding sites were identified in the list of let-7a target transcripts not bound by Sfpq, we did not plot them to compare with Figure 5A. However, we checked by RT-qPCR one of them, the Atf6B mRNA. As shown in new Supplementary Figure 7C, the let-7a-dependent downregulation of this transcript was not rescued by Sfpq knockdown. Together, these data demonstrated that Sfpq knockdown specifically upregulates let-7a targets bound by Sfpq. We have now

inserted this analysis in the "Results" section, page 10 lines 27-32 of the revised manuscript.

Fig. 6: It is expected that longer 3'UTR would have statistically higher chance to bind any RBP than short 3'UTRs. How were the data in panel C normalized to take that into account?

I am not clear about the mechanistic interpretation of presented data. The authors suggest that Sfpq aggregation facilitates let-7 recruitment, but do not provide the evidence that aggregation is required or clear interpretation how exactly Sfpq aggregation would facilitate let-7 recruitment. For example, what is different between "controlling close" and "not controlling close" Sfpq sites? ("24% and 8.2% of Sfpq-independent Ago2/let-7a peaks and endogenous Ago2/miRNA peaks, respectively, are located in the 3'UTR at close distance to Sfpq peaks (< 500 nt), suggesting that only a subset of miRNA binding sites is directly controlled by Sfpq"). For example, could Sfpq affect 3'UTR folding and affect accessibility of miRNA binding sites? The authors should make a clear distinction between cause & consequence and correlative evidences and discuss possible mechanisms of Sfpq function in discussion.

In Figure 6C, it is reported the length distribution of the 3'UTRs containing at least one Sfpq peak (green) or of those 3'UTRs containing at least one Sfpq peak together with at least one reduced-Ago2/let-7a peaks upon Sfpq knockdown (purple), we have experimentally determined by HITS-CLIP analysis. By contrast, in the grey plot we have reported the length distribution of the entire set of 3'UTRs in mouse genome. To obtain the green plot, we compared two Sfpq HITS-CLIP conditions: siControl vs siSfpq transfection of P19 cells. The computational analysis of these two HITS-CLIP conditions was performed using dCLIP program, which has been designed to perform differential analyses between two CLIP-sequencing conditions. The dCLIP program implements an MA-plot normalization method, which was originally designed for normalizing microarray data (*Smyth GK and Speed T. Methods, 31: 265-273, 2003*) and later applied to ChIP-seq analysis (*Shao Z et al. Genome Biol. 13: R16-10, 2012*). dCLIP relies on Hidden Markov mathematical/statistical model to provide the differential distribution. In our case, we considered as specific Sfpq peaks those peaks that are only present in the siControl condition whereas absent in the siSfpq condition, with a threshold score equal to 7 that summarizes the inference results of the Hidden Markov Model after collapsing neighboring bins in the same cluster with the same inference results as one region. To obtain the purple plot, we plotted the length of the 400 3'UTRs that contain Sfpq-dependent Ago2/let-7a peaks within 500 nt of distance from Sfpq peaks (Figure 4D of the revised manuscript, see "HITS-CLIP method and analysis" section of the Supplementary Methods for details).

This is the method we used to normalize the data plotted in Figure 6C. Therefore, although we agree with the Reviewer that considering a particular sequence or motif, long 3'UTRs would have statistically higher chance to bind any RBPs than short 3'UTRs, in Figure 6C we did not plot the occurrence of any sequence(s) but we have reported experimental data on Sfpq occupancy.

To provide the evidence that Sfpq aggregation promotes miRNA targeting, in the new Figures 4F and G of the revised manuscript, we have now demonstrated that the rescue of the miRNA binding activity on the 3'UTR of Lin28A, Igf2bp1, Phc3, and Phlpp2 in both nucleoplasm and cytoplasmic lysates of siSfpq-transfected P19 cells occurred with the recombinant wild-type Sfpq but not with the mutant in L535, L539, L546, and M549 substituted to alanine (Sfpq-214–598 quadruple mutant). The alanine mutations were designed to maintain the α -helical structure of Sfpq protein while disrupting the ability to aggregate, as it was previously reported (*Lee et al. Nucleic Acids Research, 43:3826-3840, 2015*). By Atomic Force Microscope (AFM), we have also validated the inability of this recombinant mutant to aggregate onto Lin28 3'UTR (Supplementary Figure 10B), although it can bind to Sfpq binding sequences (Supplementary Figure 9C). The binding ability of this mutant was also proven by Lee et al. 2015 (*Nucleic Acids Research, 43:3826-3840*). By

contrast, the control protein BSA does neither bind nor aggregate onto Lin28A 3'UTR (Supplementary Figure 10C), for details please see the answer to the last question by Reviewer 3.

The recombinant mutant Sfpq was produced from a plasmid provided by Dr. Bond (*Lee et al. Nucleic Acids Research, 43:3826-3840, 2015*). Hereafter, for the Reviewer we have provided the coomassie blue-stained gel of this protein (Figure 1 of the "Response to Reviewers"). We have now inserted these experiments in the "Results" section, page 10 lines 10-16, pages 13 lines 20-25 of the revised manuscript.

As indicated by the Reviewer, these data indicate that Sfpq aggregation may regulate 3'UTR folding to control the accessibility of selected miRNA binding sites and promote their nucleoplasmic and cytoplasmic silencing. We have now inserted this interpretation in the "Results" section, page 13 lines 31-33 and page 14 lines 5-8 in the "Discussion" section, page 15 lines 16-19 of the revised manuscript.

Figure 1.

We thank the Reviewer for the thorough and thoughtful review. We believe the Reviewer's suggestions have improved the rigor and the significance of our manuscript.

Reviewer # 2

We thank the Reviewer for his/her central question to provide more evidence about the specificity of Sfpq CLIP-sequencing data. The control experiments we performed to answer this concern has clearly further strengthened the quality of our Sfpq CLIP-sequencing data. We thank the Reviewer for the additional concerns/questions raised in the review. We hope that the new data and the changes in the revised manuscript has sufficiently strengthened the manuscript to be now suitable for publication.

Specific detailed answers follow:

[...] there are many grammatical errors, typos and unusual term usages, and so I recommend the authors to consider consulting a language editing service.

According to Reviewer's recommendation, we have now consulted a language editing service for the main text (Springer Nature Author Services) and corrected typos, grammar errors, and unusual term usages in the Supplementary Information file.

1. In the Sfpq HITS-CLIP experiment, this seems to lack important controls though this experiment is one of the major experiments in this manuscript. In Supplementary Methods (page 3), it is described that protein-RNA complexes 75-90 kD range for Sfpq were excised. But, in the corresponding Supplementary Figure 11C, the dashed boxes do not indicate the 75-90 kD, instead around 60-100kD. Which data is correct? It is essential in the HITS-CLIP experiment where are excised from the gel. Related to this point, in Supplementary Figure 11C, the authors show the data using Sfpq knockdown as a control, but it is ambiguous which bands or parts correspond to Sfpq. So, I think that the authors need to show several controls. The first is western blot data to show that the excised protein-RNA complexes contain Sfpq. The second is the data of the Sfpq-IP from cells without UV crosslink to compare with UV cross-linked samples, which is frequently used to show the specificity of the protein-RNA interactions. It is really important to represent that this Sfpq HITS-CLIP experiment was precisely performed.

The right size of the Sfpq-RNA complex excised from the membrane was around 75-110 Kd. We are sorry for the confusion that this mistake has led in the manuscript, which has been now corrected in both Supplementary Method (page 4 line 1) and new Supplementary Figure 12C (left panel). We have also corrected the Ago2-RNA complex excision to about 90-130 Kd (Supplementary Method page 4 line 1).

We have now performed the two experiment controls asked by the Reviewer. In the middle panel of the new Supplementary Figure 12C, we performed Sfpq-IP from P19 cells without UV crosslink to compare with UV cross-linked samples, whereas in the new Supplementary Figure 12D, we performed the Western blot of Sfpq-IP from UV cross-linked P19 cells upon Sfpq knockdown compared to control. These controls demonstrate that the excised membrane contains indeed Sfpq-RNA complex, but not unrelated RNAs. Therefore, this demonstrates that Sfpq HITS-CLIP experiment was correctly performed. We have now inserted these controls in the "HITS-CLIP method and analysis" section of the Supplementary Methods, page 4 line 2 of the revised manuscript.

2. In the protein-IP experiments, it is shown that the interaction between Ago2 and Sfpq is RNA-dependent. But the authors do not address what RNAs mediate this RNA-dependent interactions. The authors performed Ago2 and Sfpq HITS-CLIP, and thus would be able to examine the overlap of their target RNAs, which might mediate the RNA-dependent interaction between Ago2 and Sfpq.

According to Reviewer's suggestion, we have now overlapped the HITS-CLIP data on Ago2 and Sfpq targets (Supplementary Table 5, in the worksheets entitled "Ago2 HITS-CLIP", "Ago2-let7a HITS-CLIP" and "Sfpq HITS-CLIP"). As shown in the Supplementary Figure 6B of the revised manuscript, our data show that Ago2 binding sites mainly co-localize with Sfpq binding sites in the 3'UTR. We have now inserted this analysis in the "Results" section page 8 lines 30-31 of the revised manuscript.

3. The authors define less than 500 nucleotides as "close" in Figure 4. But 500 nucleotides are not so close enough to convince direct functional connections between Ago2 and Sfpq.

To assess 500 nucleotides (nt) as critical distance to define a direct regulation of Ago2 binding activity by Sfpq, we reasoned that within this distance, but not further, we should observe a statistically significant reduction of Ago2 peaks upon Sfpq knockdown compared to a random distribution of the distance computationally calculated. This analysis was performed in the 3'UTR as the main substrate for Sfpq-dependent Ago2 binding activity within a distance of 500 nt (Figures 4D and E). Briefly, we divided the distance between Sfpq

and the reduced Ago2/let-7a peaks upon Sfpq knockdown from 0 to 1,000 nt in bins of 250 nt each. Then, we computationally shuffled 10,000 times the relative position of Sfpq and the reduced Ago2/let-7a peaks upon Sfpq knockdown in each bin and calculated the Z score, as statistical test. The results of this analysis are shown in the new Supplementary Figure 6D, in which the solid line represents the number of reduced Ago2/let-7a peaks upon Sfpq knockdown and the box-plots the number of Ago2/let-7a dismissed peaks found after shuffling for each bin. The color in the plot indicates whether the Z score was positive (green) or negative (red). Together, this analysis indicates that (i) at closest distances between Sfpq and Sfpq-dependent Ago2/let-7a peaks the functional connection between Ago2 and Sfpq is greater and (ii) this functional connection is significant till 500 nt of distance. In conclusion, these data indicate that 500 nt is the threshold to mediate a direct functional connection between Ago2 and Sfpq. We have now inserted this analysis in the "Results" section, page 9 lines 9-22 of the revised manuscript.

4. In Figure 4F and 4G, the authors used recombinant SFPQ in the rescue experiments. But this lacks important negative controls such as unrelated proteins or SFPQ mutants. Without these negative controls, it is possible that the addition of proteins itself increases the targeting of let-7a to the targets.
5. In Figure 6, the authors claim that Sfpq aggregates in a sequence specific fashion onto long 3'UTR promote miRNA targeting. However, no evidence is shown for such a specific interaction of the Sfpq aggregates onto Lin28A 3'UTR and the aggregates of Sfpq actually facilitate the miRNA targeting. To show these points, I think it essential to take precise negative controls and perform new experiments.

To answer both questions we have now performed new experiments that provide the evidence that Sfpq aggregation is actually required to promote miRNA targeting. As shown in the new Figures 4F and G of the revised manuscript, we have now incubated either nucleoplasmic or cytoplasmic lysates from (let-7a)-siSfpq-transfected P19 cells with recombinant Sfpq mutated in L535, L539, L546, and M549 substituted to alanine (Sfpq-214–598 quadruple mutant). The alanine mutations were designed to maintain α -helical structure while disrupting the ability to aggregate, as it was previously reported (Lee *et al. Nucleic Acids Research*, 43:3826-3840, 2015). Our results indicate that while the recombinant wild-type Sfpq did rescue the downregulation of let-7 and miR-302b targeting on the 3'UTR of Lin28A, Igf2bp1, Phc3, and Phlp2, the mutant Sfpq did not.

The recombinant mutant Sfpq was produced from a plasmid provided by Dr. Bond (Lee *et al. Nucleic Acids Research*, 43:3826-3840, 2015) (Figure 1 of the "Response to Reviewers"). Importantly, as it was previously reported by Lee *et al. (Nucleic Acids Research*, 43:3826-3840, 2015), we found that the Sfpq mutant does not lose the ability to bind to RNA in a sequence specific fashion (Supplementary Figure 9C), whereas it is unable to form aggregates onto Lin28 3'UTR (Supplementary Figure 10B). By contrast, the control protein BSA does neither bind nor aggregate onto Lin28A 3'UTR (Supplementary Figure 10C), for details please see the answer to the last question by Reviewer 3.

Overall, these new data demonstrate (i) the specificity of the wild-type Sfpq protein to increase the miRNA targeting on selected binding sites and (ii) that Sfpq aggregates facilitate miRNA targeting on selected binding sites. Our experiments also indicate that Sfpq aggregation does not confer any sequence specificity, which is given by the RRM domains, as it was also previously reported (Lee *et al. Nucleic Acids Research*, 43:3826-3840, 2015; Yarosh *et al. Nucleic Acids Res.* 43:9006–9016, 2015). The sequence specific interaction of Sfpq to target RNAs was provided by HITS-CLIP, EMSA, and UV-crosslinking assays. To avoid any confusion in this matter, we have now changed the title of the last subheading in the "Results" section in "Sfpq aggregates onto target 3'UTRs to promote miRNA targeting" instead of "Sfpq aggregates in a sequence specific fashion onto target 3'UTRs to promote miRNA targeting".

We have now inserted these experiments in the "Results" section, page 10 lines 10-16, pages 13 lines 20-25 of the revised manuscript.

Minor points:

1. Many of the statistics are not correct. For example, the student t-test is not applicable to multiple samples. Please use appropriate statistics.

We have now used one-way ANOVA followed by Tukey's *post hoc* test for the analyses in Figures 4F, 4G, 5B, 5D and 5E, and Supplementary Figures 6H (for Lin28A), 7C, 7E, 8B, 8D and 8F.

2. In page 6, the authors describe “Interestingly, these proteins are known to associate with the long non-coding RNA Neat1 to form the paraspeckle, a nucleoplasmic compartment of approximately 0.2-1 um in size of not yet well understood function”. However, if you search NEAT1/Paraspeckle in PubMed, you can find many papers showing their functions in molecular and organismal levels. They should be properly cited.

We have now changed this statement and cited several references on the function of NEAT1/Paraspeckles (page 6 lines 11-15).

3. In many Figures, the labels are redundant. For example, in Figure 2 “5% input”, “IP-Sfpq”, “IP-Pspc1” and “IP-NonO” appear many times. I think it better to simplify the figures by reducing these redundancies. It would make the figures easily readable.

We have now simplified the labels in Figures 2 and 4C, and Supplementary Figures 3, 12C and D

4. In Figure 2F and Supplementary Figure 2D, the authors

We do not understand what the Reviewer meant to ask.

5. In Figure 3C, “5% Input Cytoplasm” should be corrected to “Cytoplasm”

This error has been now corrected in the revised manuscript.

6. In Supplementary Figure 2, I think it better that the labels such “Tagged Myc-Ago2” are replaced to “MYC-tagged Ago2” or just “MYC-Ago2”.

According to Reviewer's suggestion, we have now changed the label of this tagged protein to “MYC-Ago2” in both the Supplementary Figure 2 and the Supplementary Method of the revised manuscript. We have also used “Flag-Ago1” and “HA-Sfpq” in the new Supplementary Figure 2.

7. In Figure 7, RNA polymerase II does not open long range of genomic DNA region. Please illustrate it more precisely.

We have now changed the figure according to Reviewer's suggestion.

8. RNase should be corrected to RNase.

This error has been now corrected in the revised manuscript.

9. The term “Hek-293T” seems to be unusual. It should be corrected to HEK293T or another prevalent name.

We have now corrected this name throughout the manuscript

10. In page 7 “(Supplementary Table 4),,” “,” is extra.

This has been now corrected.

11. In page 11, “Interestingly, Sfpq peaks in the 3’UTR are longer...” is strange. This should be corrected.

We have now changed this sentence in the revised manuscript (page 12 line 16).

12. In page 12, “2 RRM domains bind 4 nt” is incorrect. According to reference 33, one RRM domain binds 4 nucleotides, and so 2 RRM domains recognize 8 nucleotides.

The Reviewer is right, we have now corrected this error (page 12 lines 24-25).

13. In page 12, “the motifs 4/6 nt in length” means “the motifs 4-6 nt in length”? Please specify it.

We meant 4-6 nt in length, this has been now corrected.

14. In Supplementary Methods (page 1), “NH₄HCO₃” should be corrected to “NH₄HCO₃”.

We have now changed this formula to (NH₄)HCO₃.

15. In Supplementary Methods (page 5), what is “User friendly interface website” for? Please specify this point.

We have now changed this title to “User friendly interface website for localization of the identified Ago2 and Sfpq peaks by HITS-CLIP analysis to help serve the research community”

16. In Supplementary Methods (page 8), “Gluthation-agarose” should be corrected to “Glutathione-agarose”

This has been now corrected.

We thank the Reviewer for the thorough and thoughtful review. We quite appreciate that the Reviewer’s criticisms and suggestions have led to clarifications that, we believe, have substantially improved the rigor, novelty and significance of the manuscript.

Reviewer # 3

We thank the Reviewer for his/her stated appreciation of the findings in this manuscript and for the questions he/she raised, which have provided us the opportunity to substantively enhance the quality of the manuscript.

Below, we provide a detailed response to the Reviewer.

1. The term "dismissed" and "dismissal" is used throughout (i.e. the dismissed Ago2 peaks upon Sfpq knockdown) in a way that seems odd. I am assuming the intended meaning is decreased or reduced but it is not clear. I also recommend that the term "altogether" be replaced with "together."

In the revised manuscript, we have now used “decrease” or “reduce” instead of “dismiss”, and replaced “altogether” with “together”.

2. Like wise the authors suggest a role for Sfpq in promoting "nuclear imprinting of mRNA." I feel that this is a misuse of the term "imprinting."

We made a use of this term according to a Gideon Dreyfuss' publication (*Kataoka et al., Mol Cell* 6(3):673-82, 2000). However, we agree with the Reviewer that it is rarely used in this context, therefore we have now replaced it with the more neutral term "commitment".

3. Many of the conclusions from data are over interpreted with terms like "consistently, dramatically, major and significantly" not being warranted. The discussion, however is more appropriately tempered.

We have now removed the majority of these terms in the "Results" section, while keeping those that are warranted.

4. Not sure what "we deeply analyzed..." means and figure 6A-C are over interpreted.

We have now removed "deeply" (page 12 line13). In Figures 6A-C, we have described the main features of the Sfpq binding activity found by HITS-CLIP and make a conclusion. We have now tried to temper this conclusion.

5. The annotation for significance and which bars are being referenced is unclear for most of the data represented as bar graphs (i.e. 4F and 5B).

We have now drawn new error bars.

6. Fig 4A is not represented well.

We have now replaced this Figure with a new Venn diagram.

7. The use of BSA and GST as control proteins for binding assays are not ideal. An alternative RNA-binding protein would be more appropriate.

In the new Supplementary Figure 9C, we have now performed a UV-crosslink binding assay with the recombinant Ago2. The results show no binding activity between Ago2 and the Sfpq binding sequences. We have now inserted this experiment in the "Results" section, page 12 lines 32-33 and page 13 lines 1-2 of the revised manuscript.

8. The AFM data is difficult to interpret without appropriate controls. This is not a data that will be familiar to many and It is impossible to know with any certainty that we are really looking at Lin28A 3' UTR and Sfpq aggregates. Control molecules or reference material would be helpful, especially since the Supp AFM data does not have a needed size bar. The overall size of the depicted components in Fig 6F seems questionable given that ribosomes for example are 30nm.

We used the AFM to check whether Sfpq aggregates onto Lin28A 3'UTR. Therefore as control experiment we checked whether the BSA or recombinant Sfpq mutated in L535, L539, L546, and M549 substituted to alanine (Sfpq-214–598 quadruple mutant) were unable to bind and to aggregate, respectively. The alanine mutations in the recombinant mutant Sfpq were designed to maintain α -helical structure while disrupting the ability to aggregate, as it was previously reported (*Lee et al. Nucleic Acids Research*, 43:3826-3840, 2015). Sfpq mutant was produced with a plasmid provided by Dr. Bond (*Lee et al. Nucleic Acids Research*, 43:3826-3840, 2015) (Figure 1 of the "Response to Reviewers").

From the new and old experiments, we clearly observed that by adding Lin28A 3'UTR the distribution of Sfpq proteins detected by AFM changes as follows: the recombinant wild-type Sfpq has the tendency to form clusters (or aggregates) in the presence of RNA (new

Supplementary Figure 10A, right panel), whereas the mutant Sfpq forms structures with a sort of pearl necklace shape (new Supplementary Figure 10B, right panel), indicating that the mutant Sfpq binds but does not aggregate onto Lin28 3'UTR. Moreover, Sfpq mutant does not lose the ability to bind to RNA in a sequence specific fashion (Supplementary Figure 9C). In the set of BSA visualization after incubation with RNA, we could detect neither clusters nor any change in the distribution of the protein absorbed on the surface compared to the BSA alone (Supplementary Figure 10C). This result indicates that there is no interaction between the protein and Lin28A 3'UTR. In fact, in our experimental condition (120 mM NaCl, 4 mM MgCl₂), the mica surface is not enough positively charged to allow the absorption of unbound RNA, which is removed from the surface in the washing step. Thus, Lin28A 3'UTR is only visible on the mica surface upon interaction with protein (only Sfpq in our case). In conclusion, these data demonstrate that Sfpq is able to specifically aggregate onto Lin28A 3'UTR.

We have inserted these experiments in the "Results" section, page 13 lines 20-25 of the revised manuscript, and in the Supplementary methods, page 11 lines 30-33.

We have now added the scale bars in all AFM images and corrected the scale bar in Figure 6F, which is 20 nm.

With regard to the size of our complexes visualized by AFM, it should be taken into account that *in vivo* the 3D organization of nucleoprotein complexes is a multifactorial process in which different proteins within specific environmental conditions (e.g. ionic strength) may be involved. *In vitro* the lack of this specific environmental conditions could lead to an overestimation of the complex dimensions. Indeed, previous publications reported that the dimension of nucleoprotein complexes visualized by AFM depends on the ionic strength and on the presence of molecular interactors (Krzemien *et al. PLoS One*, 12:e0173459, 2017; Hizume K, *Nucleic Acids Res.* 35:2787-99, 2007). Moreover, AFM has the tendency to overestimate the size of the molecules/structures (Vesenka J *et al. Ultramicroscopy.* 42-44 (Pt B):1243-9, 1992).

We thank the Reviewer for the suggestions, which provided us the opportunity to perform experiments and introduce qualifications that have clearly improved the rigor and the significance of the manuscript.

Summary:

We believe that we have fully addressed all concerns, questions and experiment suggestions raised by the Reviewers, and that the resulting revised manuscript is clearer and more rigorous. We hope the Reviewers will find this revised manuscript fully suitable for publication in *Nature Communications*.

Reviewers' Comments:

Reviewer #1:

Remarks to the Author:

I appreciate authors' efforts to tackle reviewer's comments, however the question about the specificity of Sfpq effects on miRNA targets is still open, although the authors seem to have all the data in hand to address it. Cumulative plots (Fig. S7B) do not provide a proper negative control. Transfection of let-7 doesn't seem to have the same destabilizing effect on its targets as in Fig S7A (possibly because the authors plotted only 54 targets), so it's hard to judge about the effect of Sfpq depletion.

The authors have genome-wide data on Sfpq and AGO2-bound RNAs (HITS-CLIP), as well RNAseq data for Sfpq KD – these should be sufficient to evaluate the specificity of Sfpq effects on RNA stability, and not only for let-7a targets. For example by plotting HITS-CLIP data (peaks/input for AGO2 and Sfpq) against RNAseq (Sfpq KD/control). What happens to shared AGO and Sfpq targets upon Sfpq KD? What happens to AGO targets without Sfpq sites under the same conditions? How do Sfpq targets without proximal AGO2 sites behave upon Sfpq KD? It would be useful for the reader to get a clear picture of how the three datasets correlate with each other, before focusing on selected bits and subsets.

Specific comment:

Fig 3B is confusing. First, AGO2 seem to be more abundant in the nucleus than in cytoplasm even in the absence of LMB, in contrary to published data. Do the authors have an explanation for that? Is the image misrepresented because of the overlapping dapi signal in magenta that is hard to distinguish from AGO2 signal? Second, AGO2 seems to be detected throughout the nucleus, so whatever is present in the nucleus would look "co-localised" with AGO2.

Reviewer #2:

Remarks to the Author:

I have reviewed the revisions and the authors appear to have made a effort to address the comments.

Point 3: I now understand that 500 nt is a critical distance to define a regulation of Ago2 binding by SFPQ. However, "direct interaction" or "direct regulation" described in the text (page 9) would be overestimation of their data, since the authors did not detect physical interactions between these factors experimentally. I recommend to make this argument milder.

Point 4: The In vitro binding experiment with the mutant SFPQ proteins (Fig. 4F,G) strengthened the author's arguments, although it is still possible that the authors picked up an artificial effect of the "sticky" SFPQ protein in vitro.

Overall, I believe the manuscript was improved and is now suitable for publication in Nature communications after minor corrections.

Reviewer #3:

Remarks to the Author:

I feel that the authors have gone to considerable lengths to accommodate most if not all of the reviewers comments. I am comfortable with the revised version of the manuscript. That said, I do not feel the final figure of the proposed model is very informative or useful. As I interpret it, it looks like although there are two populations of mRNA, Sfpq-independent and dependent (or short and long 3'UTR), their fates are essentially the same. This would seem to undermine the importance of the Sfpq control that is the basis of the manuscript.

RESPONSE TO REVIEWERS:

Here, we address the remaining concerns/suggestions of the Reviewers (here below in blue and bold), which we have completed as detailed in our response to follow.

Reviewer # 1

We thank the Reviewer for her/his comments on the specificity of the Sfpq knockdown effect on the expression regulation of miRNA targets and on the colocalization data we added in the revised manuscript. We hope that the clarifications and changes we have made in the new revised version have adequately strengthened the manuscript to be now suitable for publication.

Specific detailed answers follow:

I appreciate authors' efforts to tackle reviewer's comments, however the question about the specificity of Sfpq effects on miRNA targets is still open, although the authors seem to have all the data in hand to address it. Cumulative plots (Fig. S7B) do not provide a proper negative control. Transfection of let-7 doesn't seem to have the same destabilizing effect on its targets as in Fig S7A (possibly because the authors plotted only 54 targets), so it's hard to judge about the effect of Sfpq depletion.

The authors have genome-wide data on Sfpq and AGO2-bound RNAs (HITS-CLIP), as well RNAseq data for Sfpq KD – these should be sufficient to evaluate the specificity of Sfpq effects on RNA stability, and not only for let-7a targets. For example by plotting HITS-CLIP data (peaks/input for AGO2 and Sfpq) against RNAseq (Sfpq KD/control). What happens to shared AGO and Sfpq targets upon Sfpq KD? What happens to AGO targets without Sfpq sites under the same conditions? How do Sfpq targets without proximal AGO2 sites behave upon Sfpq KD? It would be useful for the reader to get a clear picture of how the three datasets correlate with each other, before focusing on selected bits and subsets.

We previously plotted the cumulative expression distribution of the 54 let-7a targets that do not bind to Sfpq according to HITS-CLIP data (Supplementary Figure 7B). Although let-7a in these targets does not confer a destabilizing effect as in Supplementary Figure 7A, we reasoned that if Sfpq would have any non-specific effect on these transcripts, we should have obtained a significantly different distribution upon Sfpq knockdown, independently of let-7a activity. However, we did not observe any global significant change in the expression levels between siControl and siSfpq distributions (Supplementary Figure 7B; Wilcoxon test, p value = 0.9239). Therefore, in our opinion, Sfpq knockdown specifically upregulates let-7a targets bound by Sfpq in proximity (less than 500 nt).

In both Supplementary Figure 7A and 7B we plotted in total the expression profiles of mRNAs containing either canonical (30) or non-canonical (303) and both binding sites for let-7a on 3'UTR. Non-canonical binding sites for miRNAs are known to globally confer very mild downregulation of target mRNAs compared to canonical binding sites (Agarwal, et al., 2015, Elife, 4, , Loeb, et al., 2012, Mol Cell, 48, 760-70, and Figure 5A, Supplementary Figure 7A of our manuscript). Therefore, in order to deal with more important expression changes, we focused our analysis on mRNAs that contain canonical let-7a-binding sites. Because only two

canonical let-7a-binding sites were identified in the list of let-7a target transcripts not bound by Sfpq, we did not plot them to compare with Figure 5A, which contain direct Sfpq-dependent canonical let-7-binding sites. However, we validated by RT-qPCR one of them, the Atf6B mRNA. As shown in the Supplementary Figure 7C, the let-7a-dependent downregulation of this transcript was not rescued by Sfpq knockdown. In our opinion, these data would argue in favor to a specific effect of Sfpq knockdown on the expression regulation of the mRNAs containing Sfpq-dependent let-7a-binding sites located within 500 nt of distance to Sfpq-binding sites.

As now requested by the Reviewer, to further strengthen our conclusion, we have analyzed the expression levels of all Sfpq and Ago2-bound RNAs, and not only for let-7a targets. Because, according to our HITS-CLIP data, the distance between Ago2 and Sfpq peaks in the 3'UTR matters to a direct Sfpq-dependent regulation of miRNA-binding activity (Figure 4D and E), we analyzed the expression profiles of the 118 mRNAs with a distance of less than 500 nt between Ago2 and Sfpq peaks in the 3'UTR, the 39 mRNAs with a distance between 500 and 7,000 nt between Ago2 and Sfpq peaks in the 3'UTR, and the 22 mRNAs with a distance of more than 7,000 nt between Ago2 and Sfpq peaks in the 3'UTR. These three populations of mRNAs represent the three specific datasets now requested by the Reviewer, namely, *i*) "the shared AGO and Sfpq targets" (less than 500 nt of distance between Ago2 and Sfpq peaks), *ii*) the "Sfpq targets without proximal AGO2 sites" (between 500 and 7,000 nt of distance between Ago2 and Sfpq peaks), *iii*) "the AGO targets without Sfpq sites" (more than 7,000 nt of distance between Ago2 and Sfpq peaks). With this set we should be able to see whether Sfpq knockdown specifically and globally upregulates the population of mRNAs containing direct Sfpq-dependent miRNA-binding sites (less than 500 nt between Ago2 and Sfpq peaks in the 3'UTR) and not the other two populations that contain Sfpq-independent miRNA-binding sites. Of note, in this analysis we did not consider those transcripts that only contain Sfpq-binding sites but not Ago2. In fact, we reasoned that for coherency to the other transcript populations considered in this analysis and the mechanism here studied (miRNA targeting), the transcripts containing both Ago2 and Sfpq-binding sites at a distance between 500 and 7,000 nt would fit better for this dataset.

Similar to data obtained for the regulation of mRNAs containing canonical let-7a-binding sites (Figure 5A and Supplementary Figure 7C), Sfpq knockdown significantly upregulated the steady-state expression of the 37 transcripts containing direct Sfpq-dependent Ago2 peaks with canonical binding sites for the 20 most expressed endogenous miRNAs (less than 500 nt of distance between Ago2 and Sfpq peaks) in the 3'UTR (blue line in Supplementary Figure 7D; Wilcoxon test, p value = 0.04769), but not that of the 12 and the 8 transcripts containing Ago2 peaks with canonical miRNA-binding sites located between 500 and 7,000 nt and more than 7,000 nt of distance to Sfpq peaks, respectively (Supplementary Figure 7D; Wilcoxon test, p value = 0.07729 (green line) and 0.9939 (red line), respectively).

However, none of the considered populations showed a significant upregulation upon Sfpq knockdown, when we consider the whole set of mRNAs containing Ago2 peaks for endogenous miRNAs (Supplementary Figure 7E). The failure of a significant upregulation in the latter case could be explained by the presence non-canonical miRNA-binding sites, which confer a very mild downregulation (Agarwal, et al., 2015, Elife, 4, , Loeb, et al., 2012, Mol Cell, 48, 760-70). Because of these data, the specificity of Sfpq knockdown effect on the mRNAs containing Ago2

and Sfpq peaks in proximity (less than 500 nt) can only be evaluated with mRNAs containing canonical miRNA-binding sites. Importantly, in the manuscript our findings/conclusions on the expression regulation of Sfpq-dependent mechanism of miRNA targeting are essentially based on canonical miRNA-binding sites.

Together, these data (both on the ectopic let-7a and endogenous miRNAs) correlate with HITS-CLIP analysis on the Sfpq-dependent miRNA targeting (HITS-CLIP analysis in Figure 4D and E). This correlation is clear when we consider the expression profile of target mRNAs containing Ago2 peaks with canonical miRNA-binding sites (Figure 5A, Supplemental Figure 7C, and Supplemental Figure 7D). Because target mRNAs containing non-canonical miRNA-binding sites are very mildly regulated by miRNAs (Agarwal, et al., 2015, Elife, 4, , Loeb, et al., 2012, Mol Cell, 48, 760-70), their steady-state expression levels can be globally unaffected and/or influenced by other mechanisms than miRNA targeting. Moreover, the biological significance of the non-canonical binding sites for miRNAs is much less established of that for canonical binding sites (Agarwal, et al., 2015, Elife, 4, , Loeb, et al., 2012, Mol Cell, 48, 760-70). That said, since the overexpression of let-7a simplified the data analysis and would minimize the indirect effect of Sfpq knockdown, we found that mRNAs containing Ago2 peaks with canonical or non-canonical let-7a-binding sites are specifically rescued by Sfpq knockdown only when let-7a-binding sites are located within 500 nt of distance to Sfpq peaks (Supplemental Figure 7A and B).

In conclusion, thanks to Reviewer's suggestions, we believe that we have now tackled the question about the specificity of Sfpq knockdown effects on the expression of different sets of miRNA targets according to Sfpq-binding site distance.

We have now inserted this analysis in the "Results" section, page 11 lines 9-19 (in red) of the revised manuscript.

Fig 3B is confusing. First, AGO2 seem to be more abundant in the nucleus than in cytoplasm even in the absence of LMB, in contrary to published data. Do the authors have an explanation for that? Is the image misrepresented because of the overlapping dapi signal in magenta that is hard to distinguish from AGO2 signal? Second, AGO2 seems to be detected throughout the nucleus, so whatever is present in the nucleus would look "co-localised" with AGO2.

We believe the Reviewer referred to the images for Sfpq/Ago2 colocalization. Indeed in the images for Pspc1/Ago2 colocalization Ago2 seems more cytoplasmic and its nuclear localization is not detected throughout the compartment.

For clarity, we have now changed the images of Sfpq/Ago2 colocalization (new Figure 3B). In these new images, we can clearly distinguish the cytoplasmic signal of Ago2 (in red), nuclear Ago2 (in magenta/nuances of magenta), and dapi in blue. As for Pspc1/Ago2 colocalization, we found that Ago2 has a strong cytoplasmic signal and is not detected throughout the nucleus.

Here below for the Reviewer, we have added the full field image of the new Sfpq/Ago2 colocalization images in the new Figure 3B (the white square frames in the Figure below are the new images in the Figure 3B). As the Reviewer can appreciate, the nuclear and cytoplasmic

signals of Ago2 can vary among cells (white arrows indicate some cytoplasmic localization whereas yellow arrows indicate some nuclear localization in the Ago2/dapi image). This variety may be due to technical reasons (such as, microscope focal plan) or biological reasons (such as, cell cycle phase). We would also mention that Ago2 localization and signal may vary from different cell types, as it was previously demonstrated in different publications (such as, Rudel, et al., 2008, *Rna*, 14, 1244-53) and in our data in Figure 3A and C.

Immunofluorescence

Sfpq/dapi

Ago2/dapi

Ago2/Sfpq

We thank the Reviewer for his/her scrupulous review. We believe we have addressed the remaining questions/criticisms and that the resulting manuscript has been significantly improved.

Reviewer # 2

We thank the Reviewer for her/his suggestions and the stated appreciation of our manuscript.

Below, we provide a detailed response to the remaining questions/suggestions.

Point 3: I now understand that 500 nt is a critical distance to define a regulation of Ago2 binding by SFPQ. However, "direct interaction" or "direct regulation" described in the text (page 9) would be overestimation of their data, since the authors did not detect physical interactions between these factors experimentally. I recommend to make this argument milder.

In the revised manuscript, we have changed the text at page 9 line 21 (in red) to make the statement about the "direct regulation" milder.

Point 4: The In vitro binding experiment with the mutant SFPQ proteins (Fig. 4F,G) strengthened the author's arguments, although it is still possible that the authors picked up an artificial effect of the "sticky" SFPQ protein in vitro.

It is a possibility that should have been minimized thanks to the use of the control mutant Sfpq protein, as suggested by the Reviewer.

We thank the Reviewer for the remaining suggestions and comments, which we believe, have further improved the presentation of our manuscript.

Reviewer # 3

We thank the Reviewer for the recognition of the important load of work we have done to respond the Reviewer's comments.

Below, we provide a detailed response to the remaining suggestion.

I do not feel the final figure of the proposed model is very informative or useful. As I interpret it, it looks like although there are two populations of mRNA, Sfpq-independent and dependent (or short and long 3'UTR), their fates are essentially the same. This would seem to undermine the importance of the Sfpq control that is the basis of the manuscript.

As suggested by the Reviewer, we have now changed the model to emphasize the importance of Sfpq to control a specific program of miRNA-dependent gene expression regulation.

We thank the Reviewer for his/her suggestion, which raised a serious discussion among the authors about the best way to represent our findings in a graphical abstract.

Summary:

In conclusion, we believe that we have fully addressed all the remaining questions/suggestions raised by the Reviewers, and that the resulting revised manuscript is significantly improved. We hope that all the Reviewers will now find the revised manuscript fully suitable for publication in *Nature Communications*.

Reviewers' Comments:

Reviewer #1:

Remarks to the Author:

I have no further comments on the manuscript.